# FUNDC2 promotes liver tumorigenesis by inhibiting MFN1-mediated mitochondrial fusion

Shuaifeng Li[1,2,8], Shixun Han[1,2,8], Qi Zhang[3], Yibing Zhu[1], Haitao Zhang[1,2], Junli Wang[3], Yang Zhao[1,2], Jianhui Zhao[3], Lin Su[4], Li Li[5], Dawang Zhou [6], Cunqi Ye[1], Xin-Hua Feng[1,2,7], Tingbo Liang [3] & Bin Zhao [1,2,3,7 ✉]

Mitochondria generate ATP and play regulatory roles in various cellular activities. Cancer cells often exhibit fragmented mitochondria. However, the underlying mechanism remains elusive. Here we report that a mitochondrial protein *FUN14 domain containing 2* (*FUNDC2*) is transcriptionally upregulated in primary mouse liver tumors, and in approximately 40% of human hepatocellular carcinoma (HCC). Importantly, elevated *FUNDC2* expression inversely correlates with patient survival, and its knockdown inhibits liver tumorigenesis in mice. Mechanistically, the amino-terminal region of FUNDC2 interacts with the GTPase domain of mitofusin 1 (MFN1), thus inhibits its activity in promoting fusion of outer mitochondrial membrane. As a result, loss of *FUNDC2* leads to mitochondrial elongation, decreased mitochondrial respiration, and reprogrammed cellular metabolism. These results identified a mechanism of mitochondrial fragmentation in cancer through MFN1 inhibition by FUNDC2, and suggested FUNDC2 as a potential therapeutic target of HCC.

[1] The MOE Key Laboratory of Biosystems Homeostasis & Protection, Zhejiang Provincial Key Laboratory for Cancer Molecular Cell Biology, and Innovation Center for Cell Signaling Network, Life Sciences Institute, Zhejiang University, Hangzhou 310058, China. [2] Cancer Center, Zhejiang University, Hangzhou 310058, China. [3] Department of Hepatobiliary and Pancreatic Surgery, Zhejiang Provincial Key Laboratory of Pancreatic Disease, The First Affiliated Hospital, School of Medicine, Zhejiang University, Hangzhou 310058, China. [4] Department of Ultrasound Medicine, The University of Hong Kong-Shenzhen Hospital, Shenzhen 518053, China. [5] Institute of Aging Research, Hangzhou Normal University, Hangzhou 311121, China. [6] School of Life Sciences, Xiamen University, Xiamen 361102, China. [7] Shaoxing Institute, Zhejiang University, Shaoxing 321000, China. [8] These authors contributed equally: Shuaifeng Li, Shixun Han. ✉email: binzhao@zju.edu.cn

Mitochondria are double-membrane organelles playing essential cellular functions. Besides being major sites of chemical energy production, they are also important for many cellular processes, such as biosynthesis, cell death, and innate immunity[1–3]. It is thus unsurprising that mitochondrial dysfunctions are involved in many diseases such as metabolic disorders, neurodegenerative diseases, and cancer[4,5]. Mitochondria exist as a dynamic network that constantly change morphology to maintain organelle homeostasis, and to adapt organelle functions to the extracellular environment. Mitochondrial dynamics include the movement of mitochondria along the cytoskeleton, alteration of the internal mitochondrial architecture, and connectivity mediated by fusion and fission events. Mitochondrial fusion and fission, which are relevant to mitochondrial quality control, respiration, apoptosis, and $Ca^{2+}$ homeostasis, are catalyzed by a range of proteins, especially dynamin-like GTPases. For example, mitofusin 1 and 2 (MFN1 and MFN2) *trans* associate to promote tethering and fusion of the outer mitochondrial membrane (OMM) in a GTPase-dependent manner[6,7]. Optic atrophy 1 (OPA1) is another GTPase that localizes at the inner membrane and plays an essential role in fusion of inner mitochondrial membrane (IMM)[8]. On the other hand, dynamin-related protein 1 (DRP1) is recruited into ring-like structures at the point of future fission through association with Fission 1 homolog protein (FIS1) and mitochondrial fission factor (MFF) to sever mitochondrial membranes through GTP hydrolysis[9,10]. Alternatively, FIS1 may promote mitochondrial fragmentation by inhibiting MFN1/2 and OPA1[11]. Functional significance of mitochondrial dynamics has been highlighted by genetic studies in mice demonstrating embryonic lethality upon ablation of mitochondrial fusion and fission proteins[12,13].

As the cellular power house, dysregulation of mitochondria in cancer was observed for a long time. Otto Warburg had concluded that cancer cells rely on glycolysis, and mitochondria are inactivated[14]. However, extensive research in the last decade demonstrated that most cancer cells not only maintain active mitochondria, but also derive a significant fraction of their ATP through oxidative phosphorylation[15]. In fact, an RNA interference screen had identified mitochondrial oxidative phosphorylation as the major pathway required for optimal proliferation of cancer cells in low glucose condition as that found in the tumor microenvironment[16]. Furthermore, mitochondria synthesize anabolic precursors to support rapid cell proliferation. However, cancer cells often exhibit fragmented mitochondria, as that found in lung cancer, breast cancer, and glioblastoma[17–20]. Enhanced fission or reduced fusion was linked to such a phenotype, and reversal of the phenotype by inhibition of DRP1 or overexpression of MFN2 promotes cell cycle arrest and apoptosis[21]. Mitochondrial fragmentation may promote tumorigenesis through cellular protection by isolating damaged mitochondrial portions, thus preventing catastrophe and cell death[22]. In addition, mitochondrial fragmentation was also associated with increased production of reactive oxygen species (ROS), which is a potent driver of cancer initiation, although it may have complicate roles in later stages of tumorigenesis[23]. Despite these findings, the mechanism of mitochondria fragmentation in cancer was not yet clear.

Physiological roles of mitochondrial dynamics in the liver were revealed by liver-specific conditional knockout mice. Livers of *MFN1* conditional knockout mice displayed a highly fragmented mitochondrial network, accompanied by enhanced respiration capacity, and biased use of lipid as the main energy source[24]. The mice were thus protected against insulin resistance induced by high-fat diet. Primary liver cancer (PLC) is one of the most common human malignancies and third leading cause of cancer-related mortality worldwide[25]. Hepatocellular carcinoma (HCC),

the most common form of PLC, has very limited systematic therapies, and new strategies are urgently in need. Previous studies have revealed shorter mitochondrial length in HCC tissues compared to adjacent nontumor tissues[23]. Tissue culture and tumor xenograft studies have suggested that mitochondrial fragmentation in HCC cells plays a key role in cell proliferation and migration[23,26]. However, the mechanism of dysregulated mitochondrial dynamics, and its functional roles in primary tumors are still lacking.

In this work, we systematically screened dysregulated mitochondrial proteins in HCC, and identified FUNDC2 as an elevated protein associated with poor prognosis. We demonstrated an important role of FUNDC2 in mitochondrial fragmentation by inhibiting MFN1 through physical interaction with the GTPase domain. Furthermore, reversal of mitochondrial fragmentation by *FUNDC2* knockdown reduces tumor energy level, reprograms cancer metabolism, and suppresses primary liver tumors in an MFN1-dependent manner in vivo.

## Results

**Elevated *FUNDC2* correlated with poor survival in HCC**. To systematically study dysregulated expression of mitochondrial proteins in HCC, we analyzed 1136 mitochondrial proteins from MitoCarta 3.0[27], a database of mitochondrial proteome. Transcriptional levels were examined in The Cancer Genome Atlas (TCGA) cohort of human HCC, and 267 genes were significantly upregulated in tumors (fold change > 2, $p < 0.0001$) (Fig. 1a). In order to facilitate functional studies in vivo, we examined the expression of these genes in a mouse model of primary liver tumor generated by in situ genome editing of hepatocytes[28–30]. Hydrodynamic force was generated by pressurized injection of solution into the tail vein to breach endothelium and closely associated hepatocytic plasma membrane, such that transposon plasmids were delivered into hepatocytes. Liver tumors were induced within 2–3 months after injection by transposon-mediated integration and expression of oncogenes *MYC* and *RAS* (Fig. 1b, c). Immunohistochemistry (IHC) staining of the epitope tag confirmed expression of oncogenes in tumors (Fig. 1c), and histopathological analysis revealed features of steatohepatitic HCC as previously reported[31] (Supplementary Fig. 1a). Transcriptome of tumors was profiled by RNA-seq, and filtering of dysregulated genes in $MYC + RAS$-induced mouse liver tumors identified 5 genes upregulated more than 5-fold ($p < 0.05$) (Fig. 1a). Among them, *MTHFD1L* and *GLS* have been reported to be tumor-promoting in HCC[32,33]. However, Kaplan–Meier analysis demonstrated that elevated expression of *FUNDC2* and *PRELID2* were most significantly associated with short survival of patients (Fig. 1d, Supplementary Fig. 1b–d).

Elevated mRNA levels of *FUNDC2* and *PRELID2* in HCC were validated in an independent cohort of HCC[34] (Fig. 1e). Furthermore, pairwise comparison of tumor and para-tumor tissues revealed elevated FUNDC2 protein levels in 22 out of 54 HCC samples (Fig. 1f). However, a qualified antibody for detection of endogenous PRELID2 could not be validated. Taken together, screening of genes encoding mitochondrial proteins identified overexpression of *FUNDC2* and *PRELID2* in HCC, which negatively correlated with patient survival.

**Knockdown of *FUNDC2* suppresses liver tumorigenesis in mice**. In order to determine the functional roles of upregulated *FUNDC2* and *PRELID2* in liver tumorigenesis, we first confirmed their expression levels in $MYC + RAS$ mouse liver tumors by quantitative RT-PCR (Fig. 2a). Elevated protein level of FUNDC2 was also visualized by IHC on liver sections (Fig. 2b). We then designed a multiplexed genome editing strategy, in which

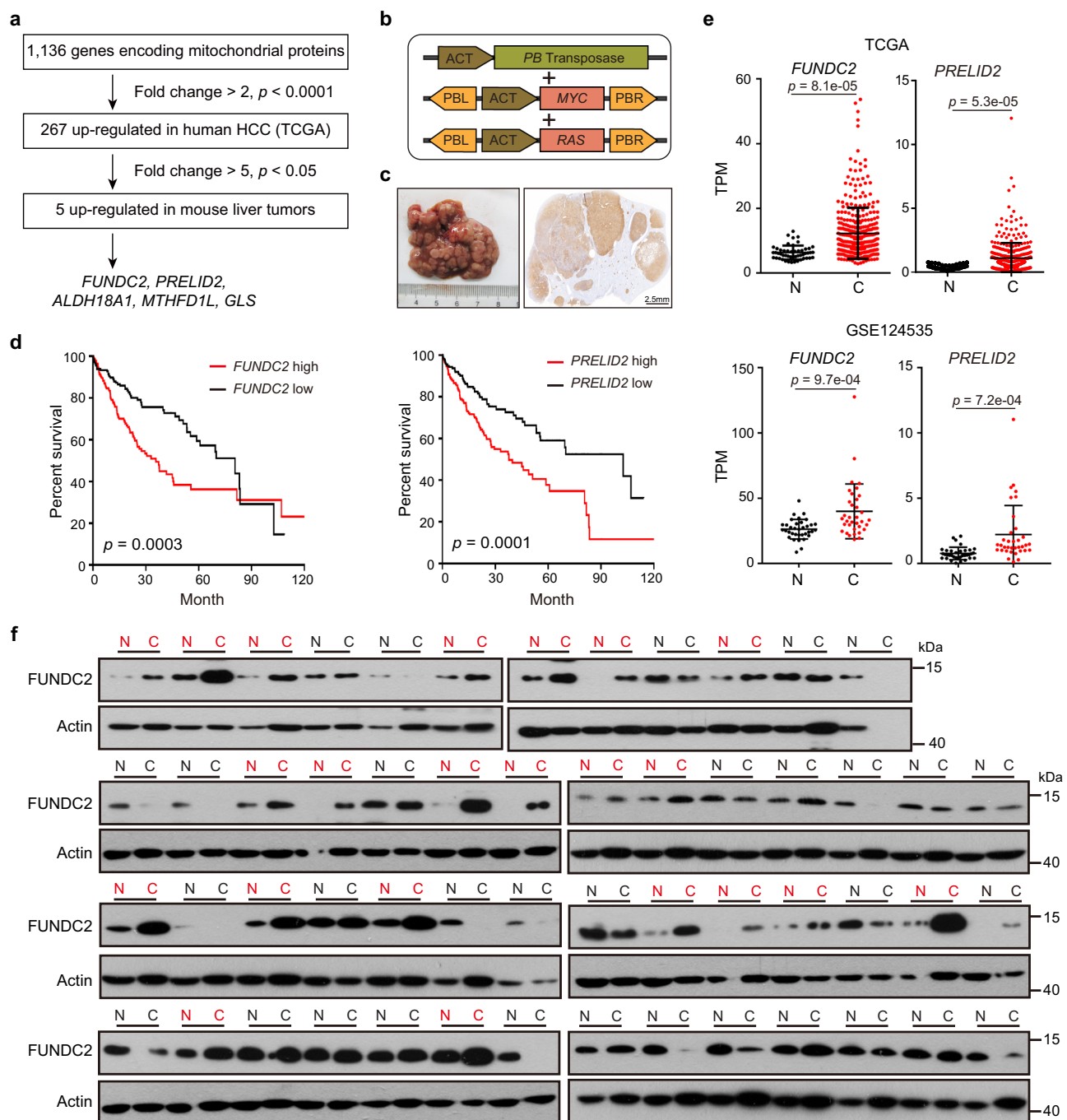

**Fig. 1 *FUNDC2* is upregulated in human HCC. a** Flowchart of screen for transcriptionally dysregulated mitochondrial protein. **b** Illustration of plasmids used for in vivo genome editing of hepatocytes. **c** Representative mouse liver tumors induced by *MYC + RAS*, and IHC staining of a respective liver section by anti-HA antibody on the right. **d** Kaplan–Meier plots of overall survival for *FUNDC2* or *PRELID2* expression levels. The median expression levels were used as a cut-off. **e** *FUNDC2* and *PRELID2* mRNA levels in the TCGA and GSE124535 cohorts of HCC. Data are presented as mean ± SD. **f** FUNDC2 protein levels in HCC (C) and adjacent normal (N) samples were revealed by western blotting. This experiment was repeated twice. Red indicates pairs with increased FUNDC2 in cancer (>2) as determined by Image J quantification of relative protein levels. *p* values were calculated by two-tailed unpaired Student's *t*-test. Source data are provided as a Source data file.

*FUNDC2*- or *PRELID2*-specific shRNAs were expressed simultaneously with *MYC* (Fig. 2c, Supplementary Fig. 2a). Knockdown of *FUNDC2* markedly suppressed tumorigenesis (Fig. 2d, e). Importantly, re-expression of *FUNDC2* rescued tumorigenesis, indicating tumor suppression was due to specific ablation of *FUNDC2* (Fig. 2d, e). However, knockdown of *PRELID2* did not affect tumorigenesis (Fig. 2f, g). Knockdown and re-expression of

*FUNDC2* and *PRELID2* in tumors were confirmed by quantitative RT-PCR (Supplementary Fig. 2b).

We further examined histopathological alterations induced by knockdown of *FUNDC2*. Markedly enhanced steatohepatitic features were revealed by hematoxylin and eosin (HE) staining in *FUNDC2* knockdown tumors, which was confirmed by oil red staining of lipid droplets (Fig. 2h, i), suggesting dysregulated

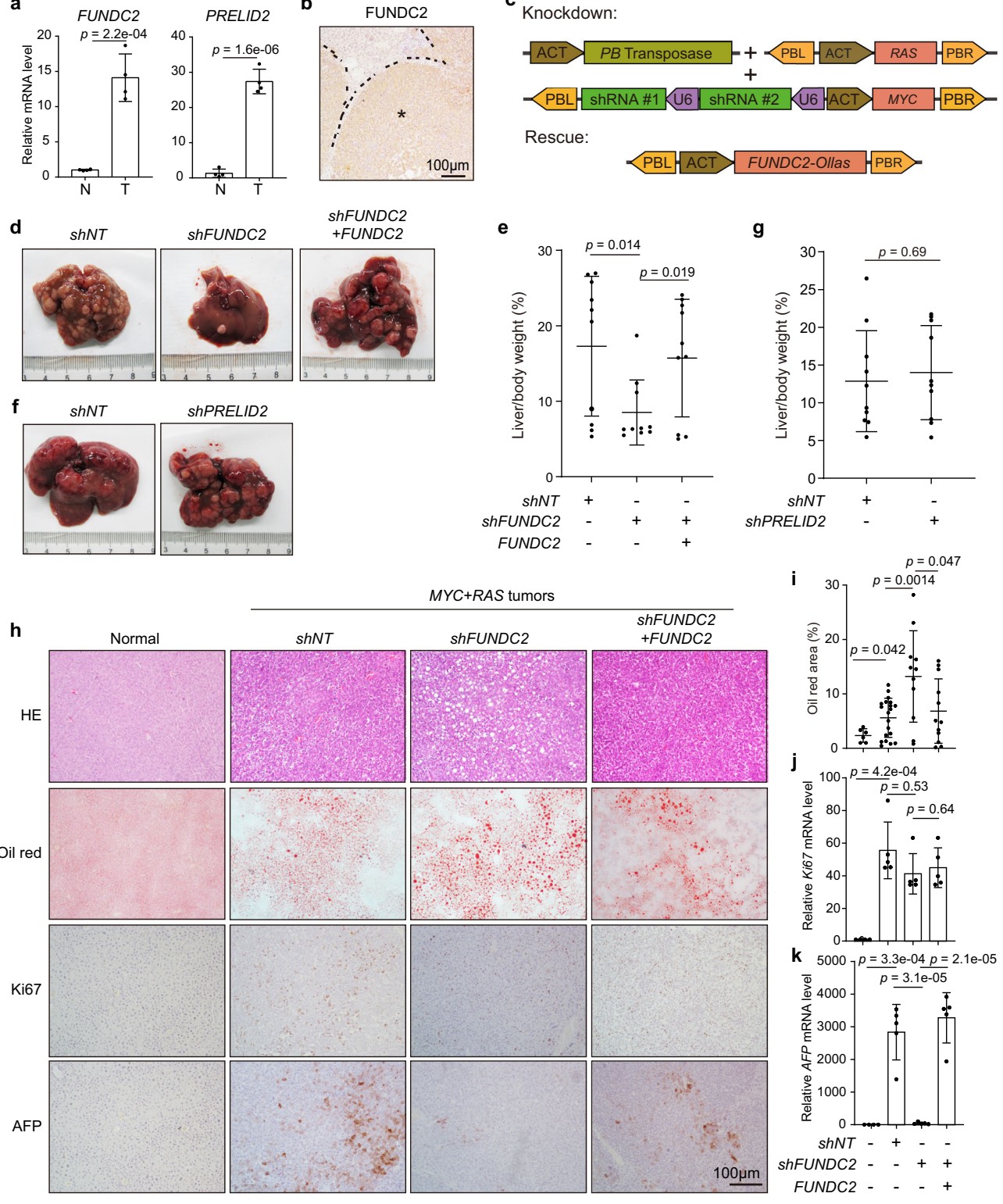

**Fig. 2 Knockdown of *FUNDC2* suppresses liver tumorigenesis. a** mRNA levels of *FUNDC2* and *PRELID2* were induced in *MYC + RAS* mouse liver tumors as quantified by RT-PCR, *n* = 4. **b** Aberrant expression of FUNDC2 protein in mouse liver tumors by IHC staining. This experiment was repeated five times. Tumor was indicated by asterisk. **c** Illustration of plasmids used for multiplexed genome editing in vivo. **d, e** Knockdown of *FUNDC2* suppressed tumorigenesis induced by *MYC + RAS*. Representative livers at 80 days after injection were shown (**d**). Liver/body weight ratios were quantified in **e**, *n* = 10. **f, g** Knockdown of *PRELID2* could not inhibit tumorigenesis induced by *MYC + RAS*. Experiments were similar to (**d, e**). **h** Impact of *FUNDC2* knockdown on histopathological features. Liver sections were stained by HE, oil red, and IHC. **i** Quantification of oil red staining from 5 sections for each group in **h**. **j, k** Determination of *Ki67* and *AFP* mRNA levels by quantitative RT-PCR, *n* = 5. n was biological replicates for all experiments. Data are presented as mean ± SD. *p* values were calculated by two-tailed unpaired Student's *t*-test. Source data are provided as a Source data file.

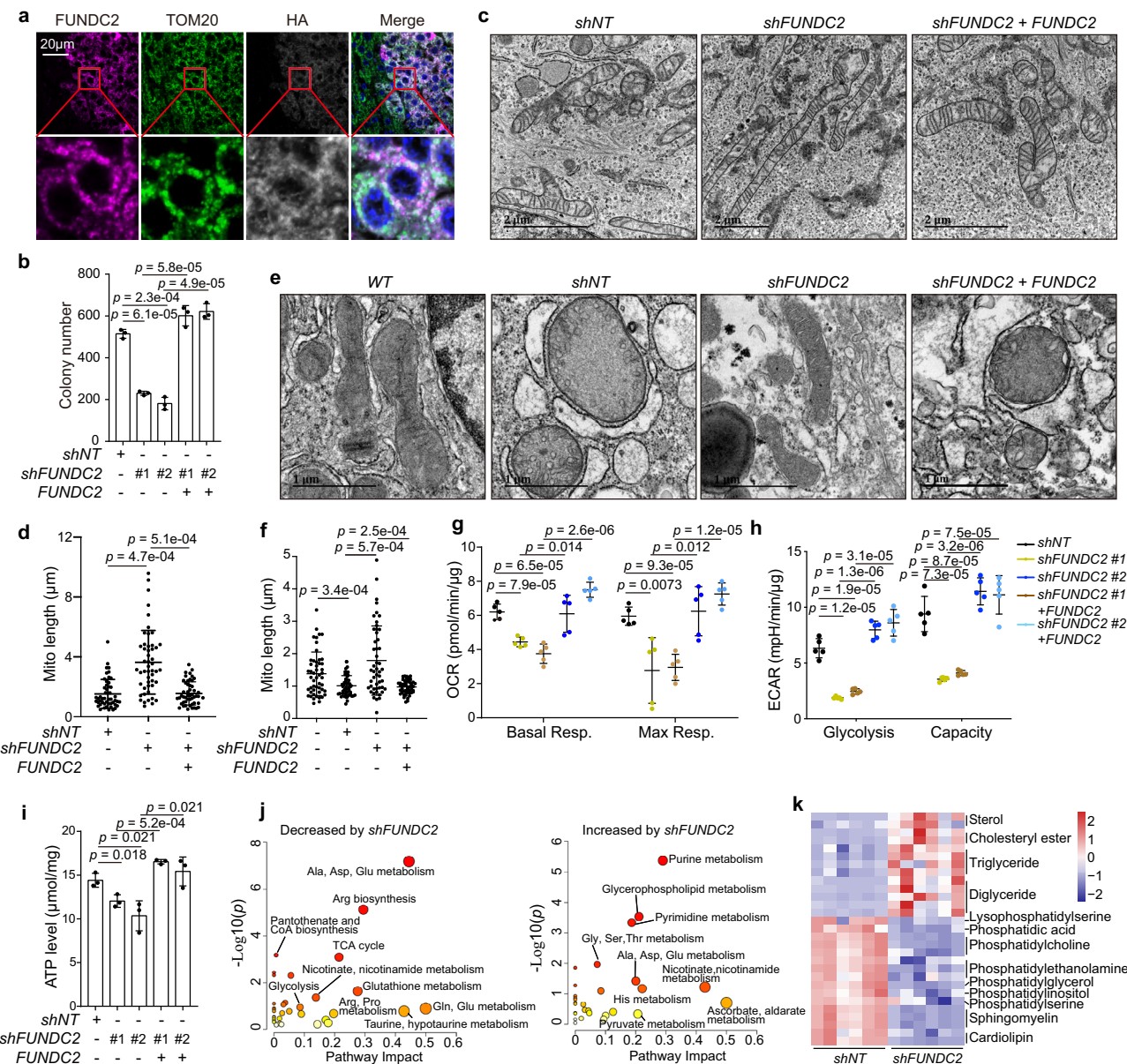

**Fig. 3 FUNDC2 promotes mitochondrial fragmentation and reprograms metabolism. a** FUNDC2 localized to mitochondria in *MYC + RAS* tumors. Multiplexed IHC staining of liver sections. Insets were enlarged for details. This experiment was repeated twice. **b** Knockdown of *FUNDC2* in Huh-7 cells reduced colony formation. Colonies were quantified by Image J, *n* = 3. **c** Mitochondrial morphology of Huh-7 cells as revealed by electron microscopy. **d** Quantification of mitochondrial length in **c**, *n* = 50 mitochondria. **e** FUNDC2 regulates mitochondrial morphology in *MYC + RAS* tumors as determined by electron microscopy. **f** Quantification of mitochondrial length in **e**, 50 mitochondria were quantified. **g**, **h** *FUNDC2* knockdown reduced mitochondrial respiration and glycolysis. Indicated Huh-7 stable cells were subjected to seahorse analysis. OCR (**g**) and ECAR (**h**) were quantified, *n* = 5. **i** *FUNDC2* knockdown reduced cellular ATP level in Huh-7 cells, *n* = 3. **j** Metabolic reprogramming in tumors by *FUNDC2* knockdown as revealed by targeted metabolomics. Pathway enrichment analysis was performed on differential metabolites. **k** Heatmap cluster analysis of lipid metabolites with differential abundance by *FUNDC2* knockdown in tumors, *n* = 6. *n* was biological replicates for all experiments. Data are presented as mean ± SD. *p* values were calculated by two-tailed unpaired Student's *t*-test. Source data are provided as a Source data file.

metabolism. Cell proliferation was evaluated by staining of Ki67. However, Ki67⁺ proliferating hepatocytes were comparable between *FUNDC2* knockdown and control tumors (Fig. 2h, j). Nevertheless, the level of AFP, which is a biomarker of HCC aggressiveness, was markedly reduced by *FUNDC2* knockdown (Fig. 2h, k). Taken together, knockdown of *FUNDC2* altered tumor cell metabolism and suppressed liver tumorigenesis in mice.

**FUNDC2 regulates mitochondrial structure and function.** We next investigated how might FUNDC2 promote tumorigenesis. It

was reported that FUNDC2 localize to the OMM[35]. We confirmed the mitochondrial localization of FUNDC2 in mouse liver tumors by multiplexed IHC (Fig. 3a). To further study the mitochondrial function of FUNDC2, we generated *FUNDC2* knockdown and re-expressing Huh-7 HCC cells (Supplementary Fig. 3a). Consistent with the pro-tumor roles of *FUNDC2* in vivo, knockdown of *FUNDC2* markedly reduced cellular colony formation on soft agar (Fig. 3b, Supplementary Fig. 3b). However, we found that mitochondrial mass, mitochondrial membrane potential, or cellular ROS levels were not affected by *FUNDC2* knockdown (Supplementary Fig. 3c–f). Nevertheless, electron

microscopy revealed that mitochondrial length was significantly longer in *FUNDC2* knockdown cells, and was rescued by re-introduction of *FUNDC2* (Fig. 3c, d). In *MYC + RAS* tumors, mitochondria were also fragmented, and abnormal swollen cristae could be observed (Fig. 3e, f). These defects were rescued by knockdown of *FUNDC2*, and were induced again by restoration of *FUNDC2* expression (Fig. 3e, f). Thus, *FUNDC2* plays an important role in mitochondrial fragmentation in tumors.

To determine the effect of *FUNDC2* knockdown on mitochondrial function, we measured oxygen consumption rates (OCR) by seahorse analysis. The results indicate that both basal and maximal respiration were reduced by *FUNDC2* knockdown, and were rescued by re-expression of *FUNDC2* (Fig. 3g, Supplementary Fig. 3g). A similar effect of *FUNDC2* knockdown on mitochondrial respiration was also observed in the HepG2 cell line (Supplementary Fig. 3h–j). By measuring extracellular acidification rates (ECAR), we found that glycolysis was also reduced by *FUNDC2* knockdown (Fig. 3h, Supplementary Fig. 3k), excluding a switch from mitochondrial respiration to glycolysis by *FUNDC2* knockdown. It should be noted that glucose uptake was not inhibited by *FUNDC2* knockdown (Supplementary Fig. 3l). Consistently, knockdown of *FUNDC2* reduced cellular ATP level, which could also be rescued by re-introduced *FUNDC2* (Fig. 3i). We further generated primary culture of *MYC + RAS* liver tumor cells (Supplementary Fig. 4a). Cellular identity was confirmed by expression of oncogenes (Supplementary Fig. 4b, c). *FUNDC2* knockdown and re-expression again inhibited and rescued mitochondrial respiration (Supplementary Fig. 4d, e), and resulted in respective changes of cellular energy level (Supplementary Fig. 4f).

By MS-based targeted metabolomics, we found that metabolites of the TCA cycle and glycolysis were decreased by *FUNDC2* knockdown in tumors (Fig. 3j, Supplementary Data 1). Importantly, these metabolites could be used for macromolecular biosynthesis, thus were believed to play important roles in supporting cancer cell growth. In contrast, purine and pyrimidine metabolites were increased. Metabolism of amino acids was both increased and decreased by *FUNDC2* knockdown. Since accumulation of lipid droplets was observed in *FUNDC2* knockdown tumors, we also carried out MS-based lipidomics, and confirmed accumulation of lipids commonly found in lipid droplets, including triglycerides, diglycerides, cholesteryl esters, and sterols (Fig. 3k, Supplementary Data 2). However, phospholipids which play critical roles in membrane formation and lipid signals fueling cell proliferation and malignancy[36], were greatly reduced by knockdown of *FUNDC2*. Taken together, elevated FUNDC2 plays important roles in energy production and metabolic reprogramming in tumor cells.

**FUNDC2 interacts with MFN1**. To determine the mechanism by which FUNDC2 regulates mitochondrial dynamics and function, we profiled FUNDC2-interacting proteins in Huh-7 cells by tandem affinity purification. Mass spectrometry (MS) revealed that about 17% of proteins co-purified with FUNDC2 were mitochondrial proteins, including MFN1 and MFN2, which are dynamin-related GTPases mediating fusion of OMM (Fig. 4a). Due to the observation that FUNDC2 regulates mitochondrial dynamics, we focused on MFN1/2. Immunoprecipitation confirmed the interaction between FUNDC2 and MFN1/2 on both ectopic and endogenous expression levels (Fig. 4b, c).

We further mapped protein sequences mediating the interaction between FUNDC2 and MFN1. FUNDC2 has its amino (N)- and carboxyl (C)-terminals exposed to the cytosol with two tandem transmembrane (TM) regions in between[35]. Truncation mutants were made according to this arrangement (Fig. 4d). Co-

immunoprecipitation experiments indicated that peptide 1–127 with deletion of the C-terminal cytosolic region retained interaction with MFN1 (Fig. 4e). However, neither the N-terminal, C-terminal, nor TM regions alone could interact with MFN1. We postulated that the TM regions may be important for OMM localization of FUNDC2, thus facilitating interaction of the N-terminal with MFN1. Indeed, addition of the TM regions resulted in mitochondrial localization of the N-terminal fragment, which was by itself diffusive in cytosol (Supplementary Fig. 5a). MFN1 also has two tandem TM regions with both ends exposing to the cytosol[37]. The larger N-terminal fragment contains a GTPase domain (Fig. 4d). Co-immunoprecipitation experiments indicated that a mutant (1–336) largely comprising the GTPase domain, and another mutant (1–584) extending more to the C-terminal could interact with FUNDC2 (Fig. 4f). It should be noted that these two mutants were cytosolic likely due to lacking of the TM regions (Supplementary Fig. 5b). Two other mutants lacking the GTPase domain could not interact with FUNDC2, although they were mitochondria-localized (Fig. 4f, Supplementary Fig. 5b). Importantly, the GTPase domain of MFN1 (1–336) could also interact with FUNDC2 1–127, indicating a direct interaction of the two regions (Fig. 4g). We next asked whether the GTPase activity of MFN1 is playing a role in interaction with FUNDC2. K88T, K222Q, and W239A mutations were reported to impair MFN1 GTPase activity[6,38,39]. Interestingly, interactions of these mutants, especially W239A, with FUNDC2 were largely reduced (Fig. 4h). Taken together, FUNDC2 is a partner of MFN1 interacting with its GTPase domain.

**FUNDC2 inhibits GTPase activity of MFN1**. Hydrolysis of GTP is critical for MFN1 *trans* association and tethering of OMM for fusion[6]. Interestingly, by an in vitro GTPase activity assay, the activity of MFN1, but not MFN2, was inhibited by co-expression of FUNDC2 (Fig. 5a). Furthermore, GTP loading of endogenous MFN1 was monitored by pulldown assay using GTP-agarose beads, which indicated that expression of FUNDC2 attenuated GTP-binding of MFN1, but not MFN2 (Fig. 5b, Supplementary Fig. 6a). Consistently, knockdown of FUNDC2 increased GTP-binding of endogenous MFN1 but not MFN2 in Huh-7 and HepG2 cells, which could be normalized by re-expression of FUNDC2 (Fig. 5c, Supplementary Fig. 6b). These data suggest that although MFN1 and MFN2 are homologous proteins both interacting with FUNDC2, they could be differentially regulated by FUNDC2.

We further asked whether FUNDC2 regulates GTP-loading of MFN1 in mouse liver tumors. GTP pulldown assay was performed using tumor lysates. While there were some variations in MFN1/2 expression levels in *MYC + RAS* tumors, GTP-binding of MFN1 was clearly enhanced by knockdown of *FUNDC2*, and repressed by restoration of *FUNDC2* (Fig. 5d). In order to determine whether FUNDC2 suppresses MFN1 GTP-binding in human HCC, we first profiled the expression of *FUNDC2* in 20 HCC samples by quantitative RT-PCR (Supplementary Fig. 6c). The top and bottom five samples were subjected to GTP pulldown assay. While slightly higher expression of MFN1 could be observed in FUNDC2-high tumors, the GTP-binding of MFN1 was clearly lower than FUNDC2-low tumors (Fig. 5e). Furthermore, co-immunoprecipitation experiments indicated that *trans* association of MFN1 could be suppressed by co-expression of FUNDC2 (Fig. 5f). Taken together, FUNDC2 inhibits the GTPase activity and *trans* association of MFN1.

**FUNDC2 regulates mitochondrial dynamics by inhibiting MFN1**. In order to determine the functional roles of MFN1 in

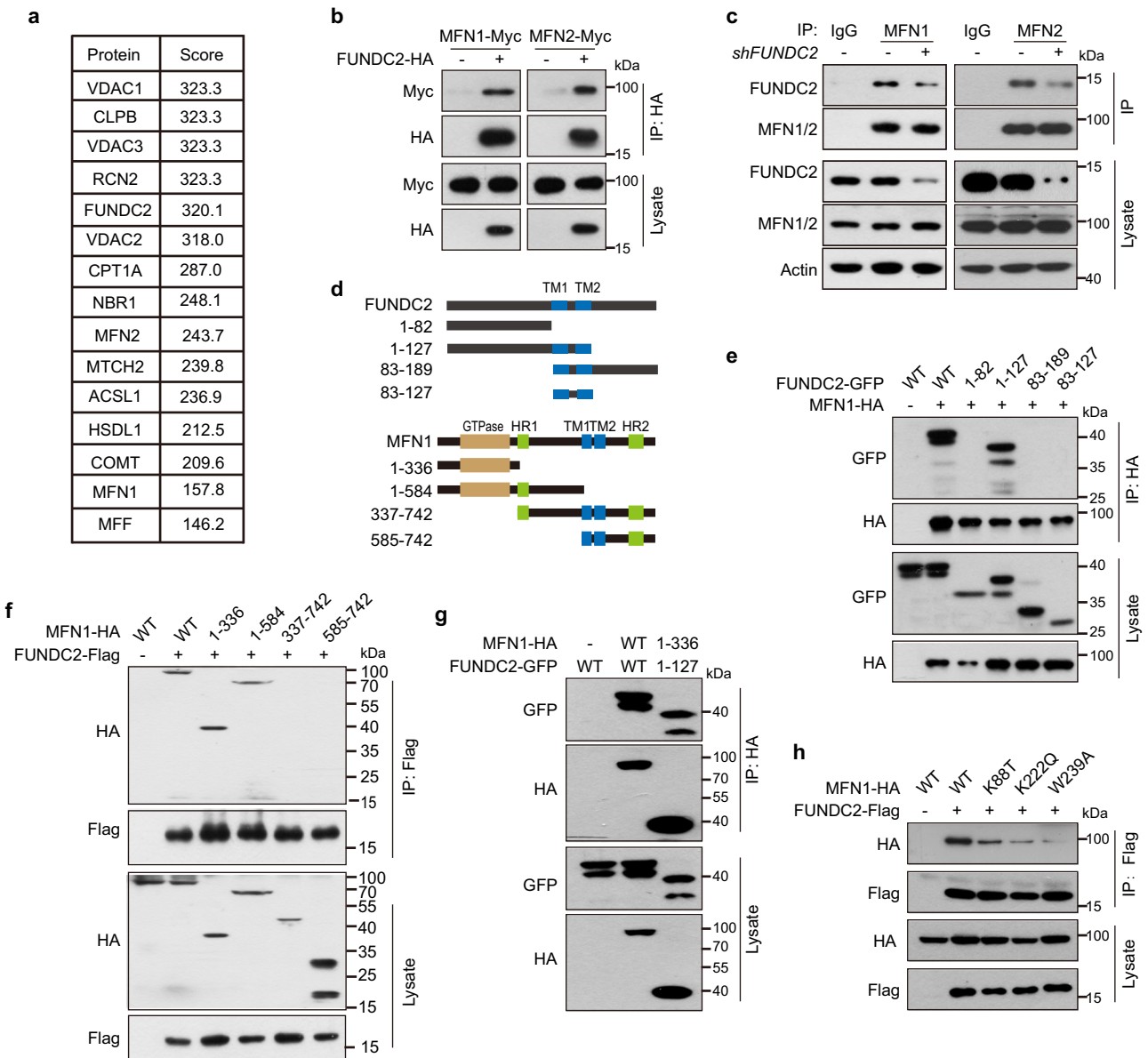

**Fig. 4 FUNDC2 interacts with MFN1. a** Identification of FUNDC2-interacting proteins by tandem affinity purification. FUNDC2-Flag-SBP was expressed in Huh-7 cells, co-purified proteins were identified by MS. Top hits were shown. **b** Confirmation of FUNDC2-MFN1/2 interaction by co-immunoprecipitation. HEK293T cells were transfected, and immunoprecipitation was done using anti-HA antibody. **c** Interaction between endogenous FUNDC2 and MFN1/2 in Huh-7 cells. **d** Illustration of FUNDC2 and MFN1 domain organization and respective mutants. Numbers indicate the position of residues. **e** Interaction between FUNDC2 mutants and MFN1 in HEK293T cells. **f** Interaction between FUNDC2 and MFN1 mutants in HEK293T cells. **g** Interaction between FUNDC2 1–127 and MFN1 1–336 in HEK293T cells. **h** Mutation of the GTPase domain impairs MFN1 interaction with FUNDC2 in HEK293T cells. Experiments in **b**–**g** were repeated three times, and **h** was repeated twice. Source data are provided as a Source data file.

regulation of mitochondria by FUNDC2, we generated *MFN1* or *MFN2* knockout Huh-7 cells by CRISPR/Cas9, which was confirmed by western blotting for MFN1 or MFN2 expression (Supplementary Fig. 7a). Cells were further infected for re-expression of *MFN1* and knockdown of *FUNDC2* (Supplementary Fig. 7a). Colony formation assay indicated that knockout of *MFN1* but not *MFN2* eliminated the effect of *FUNDC2* knockdown on reducing colony formation (Fig. 6a, Supplementary Fig. 7b). Furthermore, re-expression of wild type, but not the W239A mutant of MFN1 restored suppression of colony formation by *FUNDC2* knockdown. Consistently, Mitotracker staining revealed a requirement of wild-type MFN1, but not MFN2 for the presence of elongated mitochondria by *FUNDC2* knockdown (Fig. 6b). MFN2 has a key role in tethering

mitochondria to endoplasmic reticulum (ER), and impaired tethering could result in ER stress[40]. We thus quantified mitochondria associated ER-membranes (MAM) in tumors, and found that the percentage of MAM to mitochondria perimeter was not affected by *FUNDC2* knockdown (Supplementary Fig. 7c). In addition, knockdown of *FUNDC2* in Huh-7 cells or tumors did not induce ER stress as indicated by phosphorylation level of PERK, protein level of Bip, and protein level of ATF4 (Supplementary Fig. 7d, e).

The function of MFN1 downstream of FUNDC2 was further confirmed by a *FUNDC2/MFN1* double knockdown Huh-7 cell line (Supplementary Fig. 8a). By electron microscopy, mitochondrial elongation caused by *FUNDC2* knockdown was largely rescued by further knockdown of *MFN1* (Fig. 6c, d). Importantly,

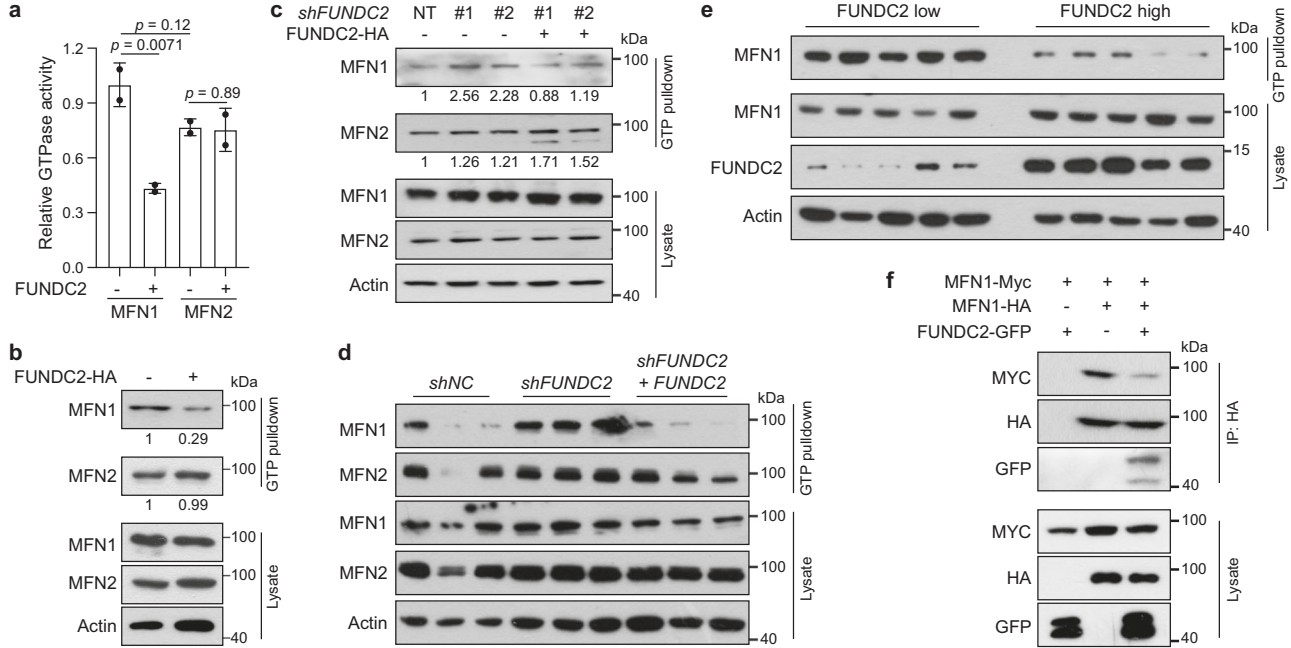

**Fig. 5 FUNDC2 inhibits GTPase activity of MFN1. a** FUNDC2 inhibits GTPase activity of MFN1 but not MFN2. MFN1/2 were expressed and immunoprecipitated from HEK293T cells, and subjected to in vitro GTPase assay, $n = 2$ biological replicates. **b, c** FUNDC2 inhibits GTP loading of MFN1. *FUNDC2* was overexpressed (**b**) or knocked down (**c**) in Huh-7 cells. Cell lysates were subjected to pulldown by GTP-agarose, and samples were then examined by western blotting. Quantification was done using Image J. **d** Knockdown of *FUNDC2* increases GTP binding of MFN1 in *MYC* + *RAS* tumors. Experiments were similar to (**b**) except that indicated tumor samples in triplicates were used. **e** MFN1 in human HCC with higher FUNDC2 levels exhibits lower binding to GTP. FUNDC2 high and low tumor samples were lysed and subjected to GTP-agarose pulldown assay. **f** FUNDC2 inhibits *trans* association of MFN1. HEK293T cells were transfected and immunoprecipitated as indicated. Data are presented as mean ± SD. *p* values were calculated by two-tailed unpaired Student's *t*-test. Experiments in **b–e** were repeated twice, and **f** was repeated three times. Source data are provided as a Source data file.

knockdown of *MFN1* blocked reduction of cellular ATP level by *FUNDC2* knockdown (Fig. 6e). We thus asked whether regulation of cellular metabolism by *FUNDC2* also depends on *MFN1*. Seahorse analysis indicated that in *MFN1* knockout cells, *shFUNDC2* could no longer reduce mitochondrial respiration, and *MFN1* wild type but not the W239A mutant could rescue the effect (Fig. 6f, Supplementary Fig. 8b). Similar observations were also made in the HepG2 cell line by double knockdown of *FUNDC2/MFN1* (Supplementary Fig. 8c, d). However, in *MFN2* knockout cells, *FUNDC2* regulated mitochondrial respiration normally (Supplementary Fig. 8e). By measuring ECAR, the regulation of glycolysis by *FUNDC2* was also found depending on *MFN1* (Fig. 6g, Supplementary Fig. 8f). Since it was reported that liver-specific knockout of *MFN1* promotes the use of lipids as energy source[24], we also measured mitochondrial respiration when palmitate was used as energy source. Surprisingly, *FUNDC2* knockdown also reduced palmitate oxidation, which was rescued by *MFN1* knockout (Fig. 6h, Supplementary Fig. 8g). This result is in consistent with the accumulation of lipid droplets in *FUNDC2* knockdown tumor cells. Taken together, inhibition of MFN1 is playing a critical role downstream of FUNDC2 in regulating mitochondrial fragmentation and respiration in tumor cells.

**FUNDC2 regulates metabolism by inhibiting MFN1.** To determine whether metabolic reprogramming downstream of FUNDC2 is also due to MFN1, we carried out MS-based targeted metabolomics comparing control and *MFN1* knockout cells. Consistent with findings by Seahorse analysis, metabolites of the TCA cycle and glycolysis were increased by knockout of *MFN1* (Fig. 7a, b, Supplementary Data 3). In contrast, pathways increased by *shFUNDC2*, including purine metabolism were

decreased by further knockout of *MFN1* (Fig. 7a, c). In order to determine how knockdown of *FUNDC2* perturbs glucose flux, cells were grown in $^{13}C_6$-glucose tracer, and incorporation of the label was analyzed by MS. Within 24 hours of tracing, *FUNDC2* knockdown cells had lower levels of M + 3 glyceraldehyde-3-phosphate, M + 3 phosphoenolpyruvate, M + 3 pyruvate and M + 3 lactate, indicating reduced glycolytic flow to lactate (Fig. 7d, e). Examining the isotopomer distribution of TCA cycle intermediates, we found a significant reduction in M + 2 citrate, M + 2 α-ketoglutarate, M + 2 fumarate, and M + 2 malate derived from $^{13}C_6$-glucose (Fig. 7f). This was in addition to an overall reduction of citrate, a-ketoglutarate, fumarate, and malate isotopomers in *FUNDC2* knockdown cells (Fig. 7f). The pentose phosphate pathway (PPP), which branches from glycolysis (Fig. 7d), is required for the synthesis of ribonucleotides, and is a major source of NADPH for the synthesis of fatty acids and the scavenging of ROS[41]. Interestingly, a significant increase in M + 6 6-phosphogluconate and M + 5 ribose-5-phosphate derived from $^{13}C_6$-glucose was observed in *FUNDC2* knockdown cells (Fig. 7g), indicating diversion of glucose to PPP. However, M + 7 sedoheptulose-7-phosphate in the revisable non-oxidative phase of PPP was reduced, suggesting that PPP was diverted to synthesize pentose phosphate from fructose-6-phosphate rather than synthesize fructose-6-phosphate from pentose phosphate (Fig. 7g). Importantly, all the above phenotypes induced by *FUNDC2* knockdown were rescued by knockout of *MFN1* (Fig. 7e–g).

In addition to accumulation of lipids in *FUNDC2* knockdown cells, targeted metabolomics revealed reduced acetylcarnitine, an indicator of mitochondrial β-oxidation. To examine changes in FAO, we fed cells with $^{13}C_{16}$-palmitate that is channeled into the mitochondria for FAO by a carrier palmitoylcarnitine (Fig. 7d).

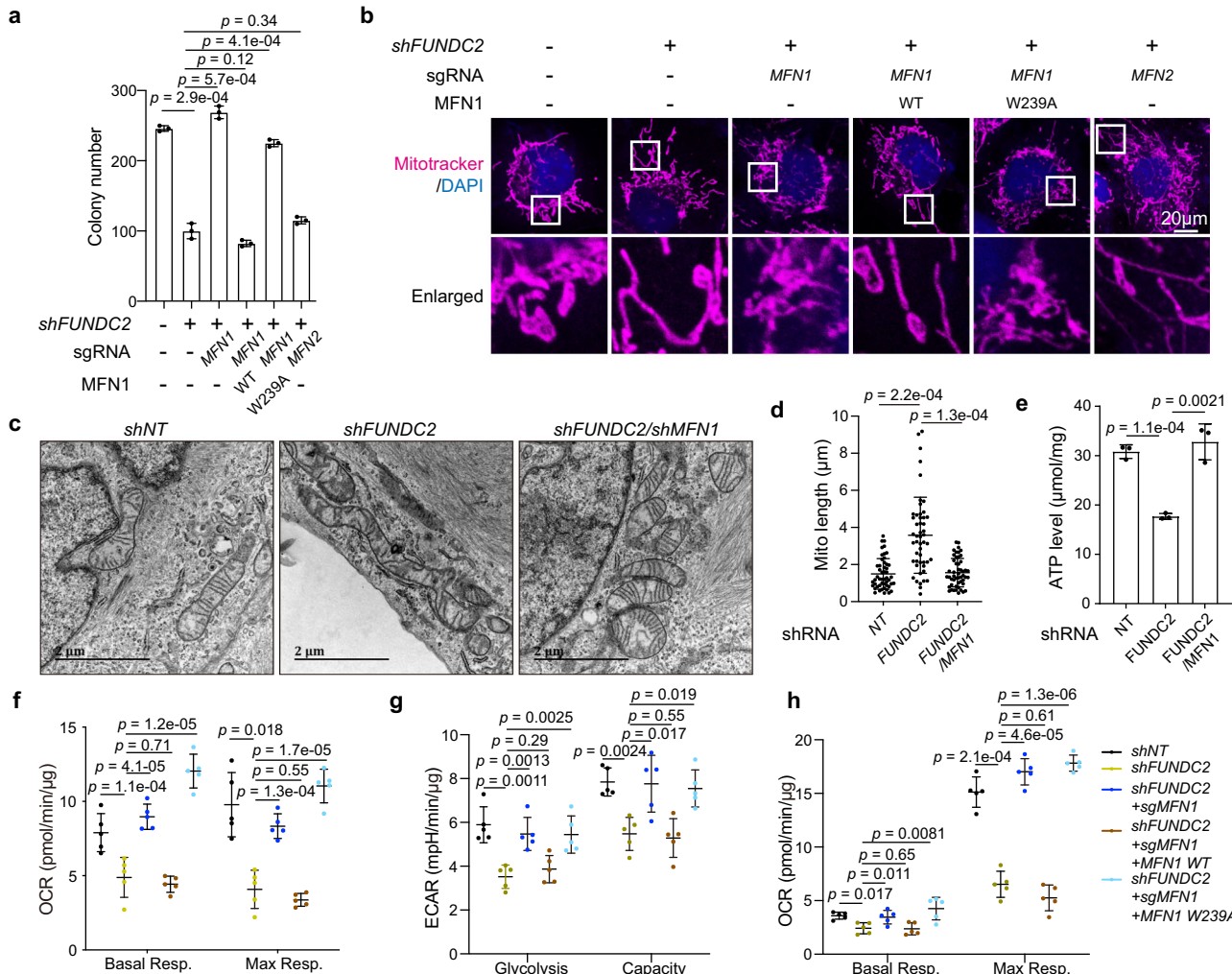

**Fig. 6 FUNDC2 regulates mitochondrial dynamics and functions by inhibiting MFN1. a** Knockout of *MFN1* rescued colony formation in *FUNDC2* knockdown Huh-7 cells. Quantification of colonies by Image J was shown, *n* = 3. **b** Mitochondrial morphology of cells in **a** was visualized by staining with mitotracker red. **c** Mitochondrial morphology as revealed by electron microscopy. **d** Quantification of mitochondrial length in **c**, 50 mitochondria were quantified. **e** Knockdown of *MFN1* rescued cellular ATP level in *FUNDC2* knockdown Huh-7 cells, *n* = 3. **f, g, h** Seahorse analysis indicated that knockout of *MFN1* rescued mitochondrial respiration (**f**), glycolysis (**g**), and fatty acid oxidation (FAO) (**h**) in indicated cells. Quantification of basal and maximal respiration (**f, h**), glycolysis and glycolytic capacity (**g**), *n* = 5. *n* was biological replicates for all experiments. Data are presented as mean ± SD. *p* values were calculated by two-tailed unpaired Student's *t*-test. Source data are provided as a Source data file.

The TCA cycle intermediates showed an overall reduction in M + 2 isotopomers, which was rescued by *MFN1* knockout (Fig. 7h). These results confirmed that FUNDC2 promotes FAO by inhibiting MFN1. Taken together, MFN1 is a critical downstream effector of FUNDC2 in reprogramming glucose and lipid metabolism in tumor cells.

**FUNDC2 promotes liver tumorigenesis via inhibition of MFN1.** If inhibition of MFN1 is a key mechanism of tumor promotion by FUNDC2, overexpression of *MFN1* should suppress tumorigenesis similar to *FUNDC2* knockdown. Indeed, co-expression of *MFN1* with *MYC* + *RAS* strongly suppressed tumorigenesis (Supplementary Fig. 9a–c). More importantly, by multiplexed genome editing, knockdown of *MFN1* abolished the tumor-suppressive function of *shFUNDC2* (Fig. 8a–c). Further co-expression of *MFN1* wild type, but not the W239A mutant rescued the tumor-suppressive function of *shFUNDC2*. Successful knockdown and rescue expression of *FUNDC2* and *MFN1* was demonstrated by quantitative RT-PCR (Supplementary Fig. 9d, e). Histopathological analysis indicated that lipid accumulation in tumors induced by *FUNDC2* knockdown was also *MFN1*-dependent (Fig. 8d, e). In addition, expression of AFP on protein and mRNA levels was also suppressed by *FUNDC2* knockdown in an *MFN1*-dependent manner (Fig. 8d, f). Thus, inhibition of MFN1 by FUNDC2 plays a critical role in liver tumorigenesis.

Consistent with reduced cellular ATP level upon *FUNDC2* knockdown in tissue culture, we found that energy level was maintained or even higher in liver tumors, but was significantly reduced by *FUNDC2* knockdown, as indicated by phosphorylated Acetyl-CoA carboxylase (ACC), a marker of energy deficiency (Fig. 8g). Furthermore, measuring of ATP level in human HCC also indicated higher energy level in FUNDC2-high tumors (Fig. 8h). Taken together, elevated expression of FUNDC2 in liver tumors not only plays an important role in mitochondrial fragmentation, but also promotes energy production to meet the demand of tumors.

**Discussion**

Reprogramming of cellular metabolism is a hallmark of cancer[15]. However, the importance of mitochondrial dysfunction in cancer

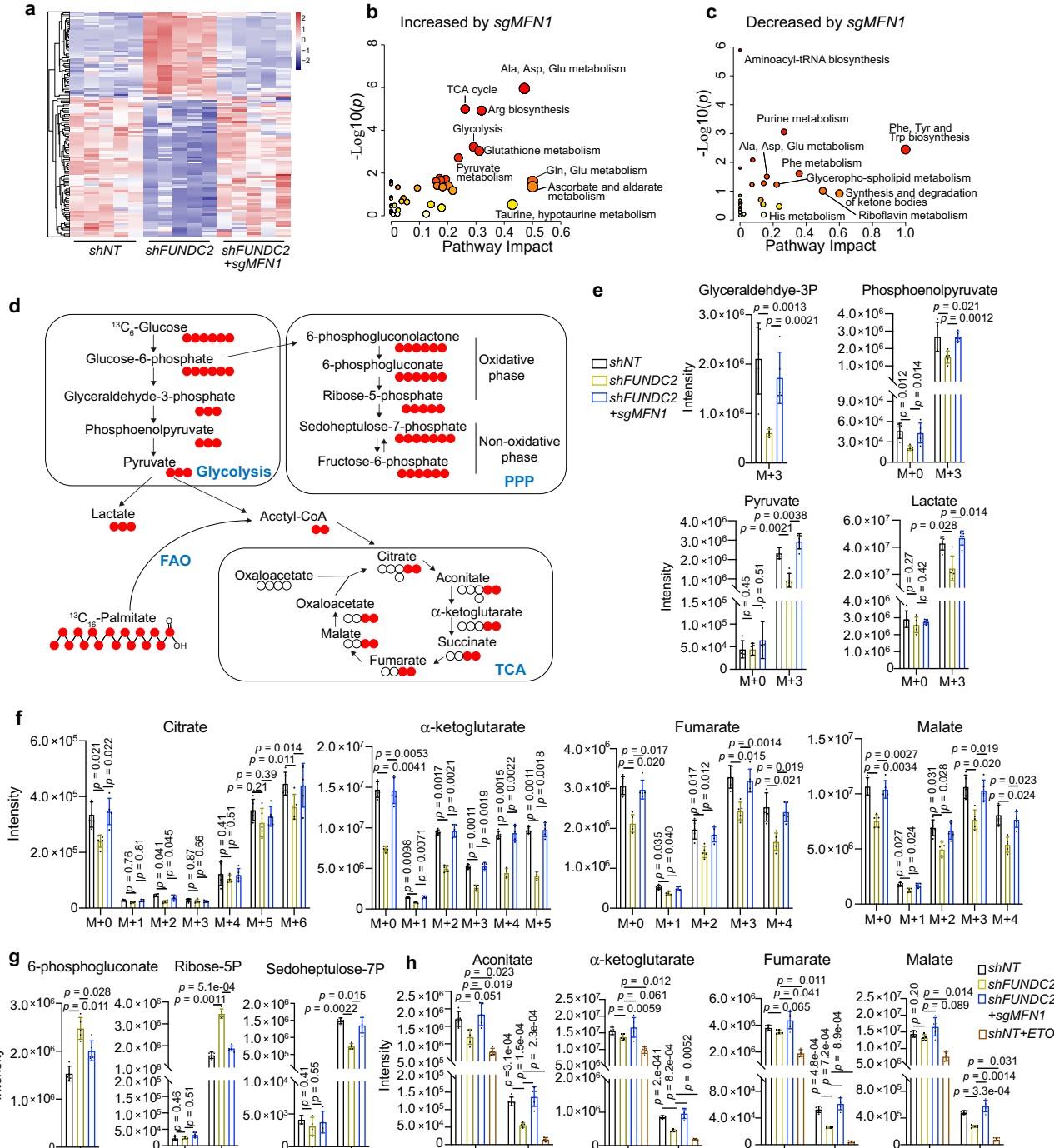

**Fig. 7 FUNDC2 regulates glucose and fatty acid metabolism by inhibiting MFN1. a** Heatmap cluster analysis of differential metabolites in Huh-7 cells with 5 biological replicates identified by metabolomics. **b**, **c** Pathway enrichment analysis on altered metabolites related to *MFN1* knockout. **d** Schematic illustration of glycolysis, TCA cycle, and PPP in ${}^{13}C_6$-glucose and ${}^{13}C_{16}$-palmitate tracing. **e**, **f**, **g** Intensity of isotopomers for glycolytic intermediates (**e**), TCA cycle intermediates (**f**), and PPP intermediates (**g**) in ${}^{13}C_6$-glucose tracing. **h** Intensity of isotopomers for TCA cycle metabolites in ${}^{13}C_{16}$-palmitate tracing. Experiments were done with 5 biological replicates. Data are presented as mean ± SD. *p* values were calculated by two-tailed unpaired Student's *t*-test. Source data are provided as a Source data file.

was overlooked until the finding that depletion of mitochondrial DNA reduced tumorigenic potential of cancer cells[42,43]. Reprogrammed mitochondrial functions support tumorigenesis in many ways. For instance, oncogenic signaling promotes the use of intermediates from the TCA cycle to generate anabolic precursors for synthesis of fatty acids and nonessential amino acids[44–46]. Excessive electron transport flux was also found in cancer cells

that not only produces ATP, but also results in formation of ROS, which plays multiple roles during tumorigenesis[47,48]. These abnormalities of mitochondria are intimately related to dysregulated mitochondrial dynamics, such that fragmented mitochondria were often found in cancer cells. Downregulation of MFN2 or elevation of DRP1 expression was found to contribute to this phenotype in lung, breast, and other cancers[49,50].

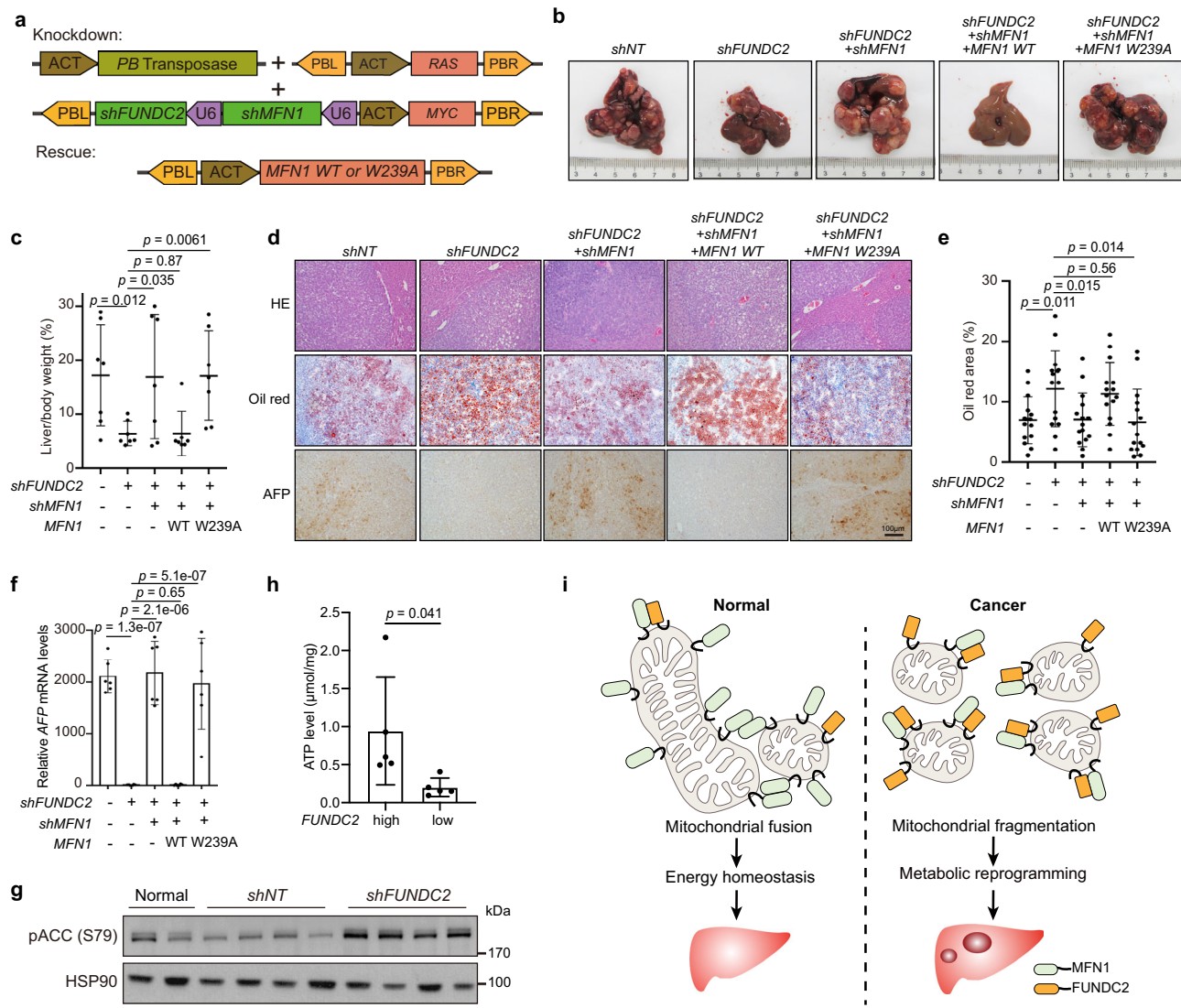

**Fig. 8 FUNDC2 promotes tumorigenesis through inhibiting MFN1. a** Illustration of plasmids used for multiplexed genome editing in vivo. **b** Knockdown of *MFN1* in addition to *FUNDC2* rescued tumorigenesis. Representative livers at 80 days after injection were shown. **c** Liver/body weight ratio was quantified, n = 7. **d** Histopathological analysis of tumors in **c**. Liver sections were stained by HE, oil red, and IHC with specific antibodies. **e** Quantification of oil red staining from 5 sections for each group in **d**. **f** Determination of *AFP* mRNA levels by quantitative RT-PCR, n = 6. **g** Knockdown of *FUNDC2* induced ACC phosphorylation in *MYC* + *RAS* tumors. Tumor lysates were examined by western blotting. **h** Human HCC with higher expression of *FUNDC2* have higher energy level. FUNDC2 high and low tumor samples were lysed and ATP levels were measured, n = 5. **i** A model depicting elevated FUNDC2 in HCC promotes tumorigenesis by inhibiting MFN1-mediated mitochondrial fusion. n was biological replicates for all experiments. Data are presented as mean ± SD. p values were calculated by two-tailed unpaired Student's t-test. Source data are provided as a Source data file.

Furthermore, post translational modifications of regulators of mitochondrial dynamics were found to play a role. For instance, MFN1 is phosphorylated at an atypical ERK site in its heptad repeat 1 domain[51]. This site was proved essential to mediate MFN1-dependent mitochondrial elongation and apoptosis downstream of the MEK/ERK cascade. However, systematic functional evaluation of dysregulated mitochondrial proteins in cancer was still absent. In this study, by screening for transcriptionally dysregulated genes encoding mitochondrial proteins in human HCC, we identified *FUNDC2* and *PRELID2*, which were correlated with patient survival. Functional characterization of these genes was facilitated by hydrodynamic injection-based multiplexed in situ genome editing of mouse hepatocytes, which achieved silencing of target genes precisely in tumor cells induced by defined oncogenes[30]. In such a way, the robust tumor-promoting function of elevated FUNDC2 was revealed. By

binding and inhibiting MFN1, FUNDC2 promotes mitochondrial fragmentation in vitro and in vivo (Fig. 8i).

Little was known about the biological functions of FUNDC2. It was demonstrated that mitochondrial FUNDC2 interacts with phosphatidylinositol-3,4,5-trisphosphate (PIP3), thereby promotes phosphorylation of AKT[35,52]. Such a function specifically supports survival of platelets. In addition, it was found that liver-specific knockout of *FUNDC2* promotes cellular accumulation of triglycerides and non-alcohol fatty liver disease (NAFLD), as well as glucose intolerance induced by high-fat diet in mice[53]. While it was suggested that FUNDC2 regulates lipid metabolism by direct binding to the promoter region of SREBP1c[54], it is in odds with the subcellular localization of FUNDC2 to mitochondria. Nevertheless, such a phenotype is consistent with our finding of reduced FAO and accumulation of lipid droplets upon ablation of *FUNDC2*. It should be noted that liver-specific knockout of

*MFN1* results in not only a highly fragmented mitochondrial network, but also enhanced mitochondrial respiration[24]. In this report, it was demonstrated that *MFN1* deficiency increased complex I abundance, which may explain increased respiration. Thus, although mitochondrial fragmentation was more thought to reduce respiration, that caused by inhibition of MFN1 seems different. Whether this difference is due to other functions of MFN1 is worth further investigation. It is also tempting to speculate that FUNDC2 may play a physiological role in the metabolism of normal liver tissue via inhibition of MFN1, and altered mitochondrial dynamics due to attenuated FUNDC2 expression may contribute to the pathogenesis of NAFLD. In addition, FUNDC2 was first cloned as an interacting partner of HCV core protein[55], which raises the possibility of dysregulated mitochondrial dynamics by HCV infection through FUNDC2. Thus, inhibition of MFN1 and mitochondrial fusion could be a major mechanism of FUNDC2 in broader biological contexts. Despite homology between MFN1 and MFN2, the later was not inhibited by FUNDC2, although physical interaction could be detected. The difference could be due to variation in critical protein sequences or differential subcellular localizations, but the exact reason awaits further investigation. In contrast to MFN1, knockout of *MFN2* in liver causes a nonalcoholic steatohepatitis (NASH)-like phenotype and liver cancer, suggesting non-complementary roles of MFN1 and MFN2[56].

It was reported that *MFN1* knockout mouse embryonic fibroblast cells were resistant to apoptotic stimuli[57]. This was due to inefficient accumulation of Bax onto OMM with incorrect curvature caused by hyperfragmentation. Whether FUNDC2 regulates apoptosis of cancer cells could be further investigated in the future. Knockdown of *FUNDC2* did not decrease the ratio of proliferating cells in tumor. However, a metabolic imbalance featuring reduced catabolic processes including glycolysis, TCA cycle, and FAO, as well as increased PPP was caused by *FUNDC2* deletion in an MFN1-dependent manner. Furthermore, although triglycerides and other storage lipids were accumulated, phospholipids which play critical roles in membrane formation and lipid signaling were reduced. Thus, by inhibiting MFN1, FUNDC2 could promote tumor growth by metabolic reprogramming following dysregulated mitochondrial dynamics. The observation that knockdown of *FUNDC2* also reduces expression of AFP, an indicator of HCC malignancy, and a marker of cancer stem cells, suggests that the cancer stem cell compartment may be more susceptible to inhibition of FUNDC2. It has been reported that in contrast to proliferating tumor cells, cancer stem cells exhibit higher dependence on oxidative phosphorylation[58,59]. In breast cancer, by distinguishing pre-existing and newly synthesized mitochondrial proteins using labeling technologies, it was found that upon asymmetric cell division, stem-like cells contained a greater number of 'new' mitochondria. Furthermore, interfering with DRP1 activity abrogated asymmetric distribution of mitochondria and reduced stem-cell properties in vitro[60]. Thus, it is possible that FUNDC2-induced mitochondrial fragmentation maintains cancer stem cells in HCC, which awaits further investigation.

In conclusion, we demonstrated MFN1 inhibition by FUNDC2 as a mechanism of mitochondrial fragmentation, which contributes to tumorigenesis of HCC. Our results suggest FUNDC2 as a potential therapeutic target of HCC.

## Methods

**Human specimens**. Human HCC specimens were collected in the First Affiliated Hospital of Zhejiang University between 2012 and 2019, who were all diagnosed with primary HCC by pathology and underwent curative surgical resection, aged 25–85, 86.8% men and 13.2% women. This study was performed in accordance with the International Ethical Guidelines for Biomedical Research Involving Human Subjects and the principles expressed in the Declaration of Helsinki, and was approved by the ethic committee of the First Affiliated Hospital of Zhejiang University. Written informed consent was acquired from the patients and the patients' parties.

**Animal model**. Animal care was provided according to regulatory standards at Zhejiang University Laboratory Animal Center. All animal study protocols were approved by the Zhejiang University Animal Care and Use Committee (ZJU20210073). Four-week-old male ICR mice were purchased from Shanghai SLAC Laboratory Animal Company. Standard laboratory chow diet for mice was purchased from XieTong Biology (Cat# 1010082), and were fed ad libitum. The SPF grade animal room was maintained humidity (45–60%) with a 12 h (7:00 a.m.–7:00 p.m.) light/dark cycle. Hydrodynamic tail-vein injection was described previously[30]. In detail, mice received 50 μg of total transposon plasmids together with 10 μg PB transposase plasmids in a volume equal to 10% of mice body weight. Plasmids for hydrodynamic tail vein injection were prepared using the Qiagen EndoFreeMaxi Kit, and were diluted in sterile Ringer's buffer before injection. Animals were euthanized 80 days after injection or when symptoms of tumorigenesis were evident, such as abdominal enlargement, lethargy or other change in behavior, such as eating, ambulation, excretion, defecate, or increased respiratory effort. Mice were euthanized by cervical dislocation or carbon dioxide. Livers were pictured and weighted, and tissues were then fixed or frozen for further processing, no data were excluded.

**Plasmid construction**. *FUNDC2*, *MFN1*, and *MFN2* were cloned into pcDNA-c-HA vector, and further sub-cloned into pLVX-c-HA, pLVX-c-Myc, and pLVX-c-Flag vectors. *FUNDC2* was also sub-cloned into pLVX-c-GFP. *FUNDC2* and *MFN1* truncation mutants were cloned into pLVX-c-GFP and pcDNA-c-HA, respectively.

shRNAs against *hFUNDC2* cDNA (*shhFUNDC2#1*: CCGGGCGTCCAGTCAAGGAAACTTTCTCGAGAAAGTTTCCTTGACTGGACGCTTTTTTG, *shhFUNDC2#2*: CCGGGATGGTGCACAGGTTTCATATCTCGAGATATGAAACCTGTGCACCATCTTTTTTG), *mFUNDC2* cDNA (*shmFUNDC2#1*: CCGGATGGTGCACTGGTTTCGTATTCTCGAGAATACGAAACCAGTGCACCATTTTTTC, *shmFUNDC2#2*: CCGGGATCTTGCAGAATTAACTAAACTCGAGTTTAGTTAATTCTGCAAGATCTTTTTC), *hMFN1* cDNA (*shhMFN1#1*: CCGGGCTCCCATTATGATTCCAATACTCGAGTATTGGAATCATAATGGGAGCTTTTTC, *shhMFN1#2*: CCGGGGCTCAAAGTTGTAAATGCTTTCTCGAGAAAGCATTTACAACTTTGAGCTTTTTC), *mMFN1* cDNA (*shmMFN1#1*: CCGGTACGGAGCTCTGTACCTTTATCTCGAGATAAAGGTACAGAGCTCCGTATTTTTC, *shmMFN1#2*: CCGGGCGAAAGAGAGAGCGTTTAAGCTCGAGCTTAAACGCTCTCTCTTTCGCTTTTTC), *mPRELID2* cDNA (*shmPRELID2#1*: CCGGAGAATGTGGTTCCAGAGATTTCTCGAGAAATCTCTGGAACCACATTCTTTTTTG, *shmPRELID2#2*: CCGGTCATTCAAACAGGCCGAATTTCTCGAGAAATTCGGCCTGTTTGAATGATTTTTG) and non-targeting (CCGGCAACAAGATGAAGAGCACCAACTCGAGTTGGTGCTCTTCATCTTGTTGTTTTTC) were expressed in pLKO.1 vector by cloning annealed oligonucleotides between EcoRI and AgeI sites.

sgRNAs against *hMFN1* (CACCGGATCTCGGAGACACATGAAGG), *hMFN2* (CACCGCCCCGTTACCACAGAAGAAC), and non-targeting (CACCGGCGGGCAGAACGACCCTGAC) were designed by the Zhang laboratory CRISPR Design Tool (https://zlab.bio/guide-design-resources). The LentiV2 vector was used to construct sgRNA-expressing lentiviral plasmids by cloning annealed oligonucleotides into BsmBI site.

**Tissue culture, transfection, and viral infection**. HepG2 (HB-8065) and HEK293T (CRL-11268) cells were from ATCC. HeLa (ATCC, CCL-2) cells were from Dr. Fangwei Wang, and Huh-7 (JCRB cell bank, JCRB0403) cells were from Dr. Junfang Ji. Huh-7, HepG2, HEK293T, and HeLa cells were cultured in Dulbecco's modified Eagle medium (Gibco, C11995500BT) containing 10% FBS and 50 μg/mL penicillin/streptomycin (Gibco, 2289322). Primary mouse liver cancer cells were cultured in DMEM/F12 (Gibco, C11330500BT) containing 10% FBS supplemented with EGF (20 ng/mL, Sigma-Aldrich, SRP3196), Insulin (5 μg/mL, YEASEN, 40112ES25) and Dexamethasone (2 μM, Sigma-Aldrich, D4902). All cells were cultured in a 37 °C humidified incubator with 5% CO$_2$. HeLa was listed as a commonly misidentified cell line by the International Cell Line Authentication Committee. We use this cell line due to its high transfection efficiency. All cell lines were authenticated by August 2021 at Genetic Testing Biotechnology Corporation (Suzhou, China) using Short Tandem Repeat (STR) analysis as described in 2012 in ANSI Standard (ASN-0002) by the ATCC Standards Development Organization. Mycoplasma test for tissue culture was done in a monthly basis using MycoPlasma Detection Kit (Vazyme, D101-01). Cells used in experiments were within 10 passages from thawing.

Transfection was performed using Lipofectamine (Thermo Fisher, L3000015) according to the manufacturer's instructions. Lentiviral or retroviral infection was used to generate stable cells. Briefly, HEK293T cells were co-transfected with viral vector and packaging plasmids, 48 h post-transfection, virus-containing medium was collected, filtered through a 0.45-μm filter and used to infect target cells in the presence of 10 μg/mL polybrene. Puromycin (Thermo Fisher, A1113803) or Blasticidin (Thermo Fisher, R21001) was used for selection.

**Colony formation assay**. For colony formation assay, cells were seeded into six-well plates at a density of 5000 cells in each well and incubated for 2 weeks, medium may be replenished every 2 days. Cells were fixed in 4% formalin for 15 min and stained with 1% crystal violet for 30 min. the number of colonies was quantified by Image J software (version 1.52r). Each experiment was repeated for three times.

**Glucose uptake assay**. 2-DG uptake was assessed using the Glucose Uptake Assay Kit (Abcam, ab136955). Briefly, Huh-7 cells were counted and plated in 96-well plate. 36 h later, cells were washed and incubated in 2% bovine serum albumin (BSA, Sigma-Aldrich, A1933) for 1 h. Then cells were stimulated by insulin, followed by 2-deoxyglucose (Sigma-Aldrich, D8375) addition for 30 min. Cells were lysed, heated at 85 °C for 40 min, and put on ice for 5 min. Then, supernatant of cell lysate was incubated with oxidation buffer and next neutralizing buffer. The plate was measured at 412 nm in a microplate reader.

**FACS analysis of mitochondria mass, membrane potential, and ROS**. For membrane potential measurement, $5 \times 10^6$ cells were incubated with 50 nM Tetramethylrhodamine methyl ester perchlorate (TMRM, Invitrogen, M20036) and 100 nM Mitotracker green (Invitrogen, M7514) for 30 min at 37 °C. For mitochondria mass assay, cells were only stained with 100 nM Mitotracker green. For CMH2DCFDA staining, $5 \times 10^6$ cells were resuspended and incubated with pre-warmed PBS containing 2 μM CMH2DCFDA (Invitrogen, C6827) for 30 min. After the incubation, cells were washed twice with PBS and analyzed by flow cytometry (Beckman Cytexpert, version 1.2). Data was processed by FlowJo (version 10).

**Western blotting and immunoprecipitation**. Western blotting was performed following standard protocol. Briefly, tissues were homogenized by tissue lysis buffer (20 mM Tris pH 7.5, 1 mM EDTA, 1 mM EGTA, 2% SDS, 150 mM NaCl, 0.1 mM DTT, 1 mM PMSF, 1 mM Na3VO4, 50 mM NaF, and protease inhibitor cocktail) and separated by SDS-PAGE and transferred onto PVDF membranes (Millipore, IPVH00010). Membranes were blocked with 5% of milk and incubated at 4 °C overnight with primary antibodies against FUNDC2 (Rabbit anti-FUNDC2, US Biological, Cat# 035793, dilution 1:5000 v/v), MFN1 (Rabbit anti-MFN1 JF0954, HUABio, Cat#ET1702-01, dilution 1:5000 v/v), MFN2 (Rabbit anti-MFN2, Proteintech, Cat# 12186-1-AP, dilution 1:5000 v/v), HNF4a (mouse anti-HNF4a H1415, Cosmo Bio, Cat# PPX-PP-H1415-00, dilution 1:5000 v/v), pACC (Rabbit anti-pACC S79, CST, Cat# 3661, dilution 1:5000 v/v), HA tag (Rabbit anti-HA C29F4, CST, Cat# 3724, dilution 1:10,000 v/v), HSP90 (Rabbit anti-HSP90, Proteintech, Cat# 13171-1-AP, dilution 1:8000 v/v), Myc tag (Mouse anti-Myc-Tag 9B11, CST, Cat# 2276, dilution 1:5000 v/v), GFP (Rabbit anti-GFP, Abcam, Cat# 6556, dilution 1:5000 v/v), β-actin (Mouse anti-ACTB 7D2C10, Proteintech, Cat# 20536-1-AP, dilution 1:10,000 v/v), ATF-4 (Rabbit anti-ATF-4 D4B8, CST, Cat# 11815, dilution 1:1000 v/v), BiP (Rabbit anti-BiP C50B12, CST, Cat# 3177, dilution 1:1000 v/v), PERK (Rabbit anti-PERK D11A8, CST, Cat# 5683, dilution 1:1000 v/v) and Phospho-PERK (Thr980) (Rabbit anti-Phospho-PERK Thr980 16F8, CST, Cat# 3179, dilution 1:1000 v/v). Membranes were then washed and incubated with HRP-conjugated secondary antibodies (Invitrogen A16096, 62-6520) for 1.5 h at room temperature. Proteins were detected using an ECL detection reagent. Scans for uncropped blots were provided in a source data file Uncropped Blots and in Supplementary Fig. 10.

For immunoprecipitation, cells were lysed by ice-cold mild lysis buffer (100 mM NaCl, 10 mM EDTA, 1% NP40, 10 mM Tris pH 7.5, 50 mM NaF, 1 mM Na3VO4, 0.1 mM DTT) supplemented with EDTA-free complete protease inhibitor (Sigma-Aldrich, S8830). Cell lysates were centrifugated at $12,000 \times g$ for 15 min at 4 °C. Supernatants were collected and incubated with desired antibodies with rotation at 4 °C for 2 h, protein G-Sepharose (GE healthcare, 17-0618-01) was added and incubated for another 1.5 h. Samples were then centrifuged, washed with mild lysis buffer for four times. Samples were boiled with 1×SDS loading buffer. Primary antibodies against MFN1 (Rabbit anti-MFN1 JF0954, HUABio, Cat#ET1702-01, dilution 1:50 v/v), MFN2 (Rabbit anti-MFN2, Proteintech, Cat# 12186-1-AP, dilution 1:50 v/v), HA tag (Rabbit anti-HA C29F4, CST, Cat# 3724, dilution 1:200 v/v) and Flag tag (Mouse anti-Flag M2, Sigma-Aldrich, Cat# A8592, dilution 1:100 v/v) were involved in this section.

**Real-time qPCR**. Total cellular RNA was extracted with Trizol (TaKaRa, Cat# 9109). Subsequently, complementary DNAs (cDNAs) were synthesized using First-Strand Synthesis System (TaKaRa, Cat# RR036A) according to the manufacturer's instructions. cDNA was analyzed by qPCR with SYBR Green (YEASEN, Cat# 11143ES50) and gene-specific primers (mPRELID2, forward 5′-GTCGCTTGC TTCCTC-3′, reverse 5′-GATTTTCTACGCTTTCC-3′; mFUNDC2, forward 5′-GCTAACAGTCAAGGAAA-3′, reverse 5′-TCTGGAATACGAAACC-3′; mMFN1, forward 5′-ATCACTGCAATCTTCGGCCA-3′, reverse 5′-AGCAGTTGG TTGTGTGACCA-3′; mSDHA, forward 5′-GAAGATTTATCAGCGTG-3′, reverse 5′-GTGTAAGAGTGAGTGGC-3′; mKi67, forward 5′-GCTCACCTGGTC ACCATCAA-3′, reverse 5′-TGACACTACAGGCAGCTGGA-3′; mAFP, forward 5′-GTTTCCAGAACCTGCCGAGA-3′, reverse 5′-CTGAGCAGCCAAGGACAG AA-3′; hHPRT, forward 5′-AGCCCTGGCGTCGTGATTA-3′, reverse 5′-ACAA TGTGATGGCCTCCCA-3′; hFUNDC2, forward 5′-ACTGGCAACGAGTGGA-GAAG-3′, reverse 5′-CATGCCAAGCAGAAAGCCTC-3′. Relative expression of mRNA was normalized by *Succinate dehydrogenase, subunit A* (*Sdha*) or *hypoxanthine phosphoribosyltransferase 1* (*HPRT1*) mRNA. The real-time PCR results were analyzed and expressed as relative expression of CT (threshold cycle) using the $2^{-\Delta\Delta Ct}$ method.

**GTP-binding and GTPase activity assays**. GTP-binding assay was performed as previously reported[61]. Briefly, 100,000 cells were lysed by lysis buffer (100 mM NaCl, 10 mM EDTA, 1% NP40, 10 mM Tris pH 7.5, 10 mM MgCl2, 50 mM NaF, 1 mM Na3VO4, 0.1 mM DTT). Lysates were centrifuged for 10 min at $12,000 \times g$, and supernatant aliquots were used to determine total protein levels. Lysates were then incubated with GTP-agarose suspension (Sigma-Aldrich, G9768) for 1 h at 30 °C with agitation. Agarose beads were collected by centrifugation, washed 3 times in lysis buffer, and resuspended in 40 μl SDS-PAGE sample buffer. GTP-bound proteins were analyzed by immunoblotting.

For measurement of MFN1 and MFN2 GTPase activity, MFN1-HA and MFN2-HA were immunoprecipitated from cell lysates, and GTPase activity was measured with a GTPase Activity kit (Sigma-Aldrich, MAK113) according to the manufacturer's instructions.

**Immunofluorescent staining and imaging**. Cells were seeded and transfected on glass coverslips, 24 h post-transfection, cells were fixed with 4% paraformaldehyde in PBS for 15 min, permeabilized with 0.1% Triton X-100 for 5 min. cells were blocked and incubated with primary antibodies against HA tag (Rabbit anti-HA C29F4, CST, Cat# 3724, dilution 1:500 v/v) and OLLAS tag (Rat anti-OLLAS L2, Novus Biologicals, Cat# NBP1-06713, dilution 1:100 v/v) for 2 h at room temperature. After washes, cells were stained with indicated fluorophore-conjugated secondary antibodies (Thermo Fisher, A-11008, A-11006) for another 1 h at room temperature. After wash, coverslips were mounted by ProLong Gold antifade mounting media with DAPI (Thermo Fisher, P36941). Images were taken by an LSM 880 confocal microscope (ZEISS).

**Transmission electron microscopy**. Briefly, tumors were cut into pieces of about 1 mm³, and cells were grown on coverslips. Samples were fixed with 2.5% glutaraldehyde (Sigma-Aldrich, G5882) for 2 h at room temperature. After wash, samples were post-fixed with 1% osmium tetroxide (Sigma-Aldrich, O5500) for another 1.5 h, and dehydrated with an ethanol series. Samples were infiltrated, embedded in Epon Resin, and polymerized at 60 °C for 12 h. Ultrathin sections of 60 nm were prepared, stained with uranyl acetate and lead citrate. Samples were observed under a Hitachi HT7700 transmission electron microscope. Mitochondrial length was analyzed by an Image-Pro Plus 6.0 software. For quantification of MAM, we normalized the MAM region to total mitochondrial perimeter following a published method[62].

**Seahorse assay**. Cellular oxygen consumption rates (OCR) were measured in real time using the Seahorse XF 96 Extracellular Flux Analyzer (Seahorse Bioscience). Briefly, 10,000 cells were seeded into 96-well Seahorse microplates in 100 μL growth medium (Agilent, 103015-100) and incubated at 37 °C in 5% CO2 overnight and the calibrator plate was equilibrated overnight in a non-CO2 incubator. Before the test, cells were washed twice with assay running media (unbuffered DMEM, 25 mM glucose, 1 mM glutamine, 1 mM sodium pyruvate) and incubated for 1 h in a non-CO2 incubator. Once the probe calibration was completed, the probe plate was replaced by cell plate. Cellular OCRs were measured by injection of the following compounds: 1 μM oligomycin (MedChemExpress, HY-N6782), 0.5 μM FCCP (MedChemExpress, HY-100410), and 1 μM antimycin A (MedChemExpress, HY-107406) plus rotenone (MedChemExpress, HY-B1756). At the conclusion of the assay, cells were lysed and protein levels were measurement by BCA kit. OCR was normalized by the amount of total protein.

For glycolysis stress test, cells were incubated with glucose-free medium supplemented with 1 mM pyruvate at 37 °C in incubator without CO2 for 1 h prior to the assay. Glucose, oligomycin and 2-DG were injected into plates in order. Extracellular acidification rate (ECAR) was normalized by the amount of total protein.

Palmitate-BSA was prepared using 2 mM palmitate solution heated at 70 °C, and 0.1 M NaOH was added until the solution was clear. To keep the ratio of palmitate to BSA 5:1, palmitate solution was added to 0.34 mM BSA solution (0.9% NaCl, 65 °C) drop by drop until the solution was clear. Finally, the 1 mM palmitate-BSA solution was filtered and stored at −80 °C. For fatty acid oxidation assay, cells were incubated overnight with seahorse substrate-limited medium with 0.5 mM glucose, 1× GlutaMAX (Gibco, 35050061), 0.5 mM carnitine (MedChemExpress, HY-B0399) and 1% FBS. Before the assay, cells were replenished with FAO assay medium (Agilent, 103693-100) and incubated for 30 min. After that, cells were pre-treated with etomoxir (40 μM, MedChemExpress, HY-50202), palmitate (200 μM) or BSA for another 20 min. Oligomycin, FCCP, antimycin A plus rotenone were injected into the plate in order. OCR was analyzed and normalized by the amount of total protein.

**Measurement of cellular ATP level**. ATP was measured using an ATP Assay Kit (Beyotime Biotechnology, S0026) according to the manufacturer's instructions. Briefly, cells were homogenized in ice-cold lysis buffer. After centrifugation, supernatants were added into substrate solution and the luminescence was recorded by a microplate reader (POLARstar Omega). A standard curve of ATP concentration was prepared from measurement of standard solutions. Protein concentration was determined by the BCA method for normalization of ATP concentration.

**Tandem affinity purification**. To identify FUNDC2-interacting proteins, Huh-7 cells stably expressing FUNDC2-Flag-SBP were generated. Cells were lysed with lysis buffer (10 mM Tris pH 7.5, 2 mM EDTA, 150 mM NaCl, 0.3% CHAPS, 2.5 mM NaF, 1 mM $Na_3VO_4$, 0.1 mM DTT and 0.5 mM PMSF) supplemented with EDTA-free complete protease inhibitor. Lysates were centrifuged at $12,000 \times g$ for 15 min at 4 °C, and supernatants were collected and incubated with anti-Flag M2 resin (Sigma-Aldrich, F2426) for 2 h at 4 °C. Resins were collected by centrifugation and washed three times with lysis buffer. Bound proteins were eluted with 200 ng/μL 3×Flag peptide (Sigma-Aldrich, F4799). Eluates were further incubated with streptavidin-conjugated resin (Agilent, 240105) for another 2 h at 4 °C. Then resins were washed and eluted with elution buffer (Tris pH 7.5, 150 mM NaCl, 0.05% Rapigest, 10 mM 2-mercaptoethanol, 4 mM biotin). Samples were then analyzed by MS/MS.

**IHC staining**. Mouse livers were fixed in neutral buffered formalin for 24 h at room temperature and then embedded and processed according to standard protocols. Liver sections were deparaffinized through graded ethanol solutions. After an antigen retrieval procedure of 30 min, sections were stained with specific antibodies using the avidin-biotin complex system (Vector Laboratories, SP-2001, PK-6100). 3,3'-diaminobenzidine (DAB, Vector Laboratories, SP-4105) was used as the substrate. Cell nuclei were counterstained with hematoxylin. Primary antibodies against AFP (Rabbit anti-AFP, Abcam, Cat# ab46799, dilution 1:500 v/v) and Ki67 (mouse anti-Ki67 B56, BD, Cat# 556003, dilution 1:500 v/v) were involved in this section.

**Histopathological analysis**. For histopathological analysis, HE staining was performed on paraffin-embedded tissues. Lipid droplets were visualized by Oil Red (Servicebio, G1015) staining of Tissue-Tek O.C.T compound-embedded frozen liver sections. Histopathological images were captured under a light microscope. Images were quantified using Image J software (version 1.52r).

**Fluorescent multiplexed immunohistochemistry**. Fluorescent multiplexed immunohistochemistry was performed with Opal 7-color Manual IHC Kit (AKOYA Biosciences, NEL811001KT) according to the manufacturer's protocol. In brief, sections were deparaffinized, microwave treated in epitope retrieval buffer for 45 s at 100% power and an additional 15 min at 20% power, blocked in Opal Antibody Diluent/Block at room temperature for 10 min, incubated with the specific primary antibody overnight at 4 °C, 10 min with the secondary horseradish peroxidase-conjugated antibody Polymer HRP Ms + Rb at room temperature, and 10 min with Opal fluorophore working solution. Sections were rinsed between staining steps with 1×Tris buffered Saline with Tween 20 and stripped between each round of staining via microwave treatment in antigen retrieval buffer. After the final microwave treatment, slides were stained with DAPI for 10 min followed by mounting. Images were acquired with confocal microscope LSM 880 (Zeiss).

**Metabolite extraction and quantitation**. Intracellular metabolites were extracted using a method described previously[63,64] with modifications. Cells were quenched using 60% methanol pre-cooled at −80 °C after a brief wash with pre-warmed PBS, then lysed with 5 freeze-and-thaw cycles. Soluble metabolites were separated by two rounds of centrifugation and dried with a CentriVap Concentrator system (Labconco). For extracting tissue metabolites, ~5 mg of tissue was homogenized and lysed with bead-beading at 4 °C in MS-grade methanol, followed by 2 rounds of centrifugation at $15,000 \times g$ for 15 min at 4 °C. The supernatant was dried in a CentriVap Concentrator system (Labconco), and the pellet was collected to measure protein concentration and subsequent normalization for injection. Dried metabolite extracts were resuspended in 60% acetonitrile and injected for quantification a triple quadrupole mass spectrometer (the QTRAP 6500+ System, ABSCIEX) coupled with an ultrahigh performance liquid chromatography. Metabolites were separated chromatographically on a SeQuant Zic-pHILIC column (5 μm polymer 150 × 2.1 mm, Millipore Sigma) and monitored with corresponding MRM transitions established using chemical standards. A 34-min liquid chromatography program running at a flow rate of 0.15 mL/min was used. In detail, Buffer A: 20 mM ammonium carbonate and 0.1% (v/v) ammonium hydroxide, and Buffer B: acetonitrile. $T = 0$ min, 80% B; $T = 20$ min, 20% B; $T = 20.5$ min, 20% B; $T = 34$ min, 80% B. Metabolites detected by MRM transitions in both positive and negative modes were carefully re-inspected for accuracy. The area under each peak was quantitated using a SCIEX OS software (version 1.7).

**$^{13}C_6$ glucose and $^{13}C_{16}$ palmitate tracing**. [U-$^{13}$C] glucose (Sigma-Aldrich, 389374) and [U-$^{13}$C] palmitate (Sigma-Aldrich, 605573) were used to trace TCA cycle metabolites. Cells were seeded into 60 mm dish, after 24 h, the medium was replaced with conditional DMEM containing [U-$^{13}$C] glucose or [U-$^{13}$C] palmitate with 10% dialyzed FBS. Cells were traced with [U-$^{13}$C] glucose (25 mM) for 24 h or with [U-$^{13}$C] palmitate (200 μM) for 36 h. $^{13}$C incorporation into TCA cycle metabolites leads to various forms of mass shift. We established a method to examine these $^{13}$C-labeled metabolites in the negative ion mode according to a previous study[65] with minor modifications. The MRM transitions for detecting $^{13}$C-labeled TCA cycle metabolites were verified using metabolites extracted from cells grown with [U-$^{12}$C] and [U-$^{13}$C] glucose. The tracing duration and labeling efficiency are also optimized.

**Lipid extraction and quantitation**. Tissue lipids were extracted with a previous method[66] with some modifications. A mixture of 17:0 PC, 17:0 PE internal standards was added to each 10 mg tissue sample and then lysed by bead-beading at 4 °C in MS-grade methanol. Cell pellets and lysates were then transferred to glass tubes, and chloroform was added. After a vigorous vortex, samples were centrifuged, and the supernatant was transferred to a new glass tube. Pellets were saved for measuring protein concentration using a BCA assay. Chloroform and citric acid were added, vortexed, and phase-separated by centrifugation. The bottom lipid phase was harvested and dried using a CentriVap Concentrator system (Labconco). Lipid extracts were reconstituted in a sampling buffer consisting of isopropanol: acetonitrile: water (2:1:1, v/v/v) and injected for a non-targeted lipidomics analysis using a SCIEX QTOF 6600+ System with a scan range of $m/z$ 100–1500 in both positive and negative ion modes. For better resolution, lipid samples were separated chromatographically using the LC system coupled to the QTOF system. We ran the samples under a 17-min program on an ACQUITY UPLC BEH C18 column (Waters, 130 A, 1.7 μm, 2.1 mm × 50 mm) at a flow rate of 0.15 mL/min. Solvent A consisted of methanol/acetonitrile/water (1:1:1, v/v/v) and solvent B was isopropanol, both solvents containing 5 mM ammonium acetate. The gradient started with 20% B for 1 min, steadily increased to 60% B by 3 min, reached 98% B by 13.1 min, and then decreased to 20% B by 16 min. Raw data files were processed using the MS-DIAL software (version 4.6) to identify and quantify lipid species[67].

**Statistics and reproducibility**. GraphPad Prism was used for graphical representation and statistical analysis of data. Data are presented as mean ± SD. No statistical methods were used to estimate sample size. A standard two-tailed unpaired Student's $t$ test was used for statistical analysis of the two groups. All experiments were repeated three times unless specified. All shown results were consistent among replicates. Kaplan–Meier survival analysis was used to estimate overall survival.

**Reporting summary**. Further information on research design is available in the Nature Research Reporting Summary linked to this article.

## Data availability
All data are available in the main text or supplementary materials. Human data derived from the TCGA and GSE124535 datasets are available from https://portal.gdc.cancer.gov/legacy-archive/search/f and https://www.ncbi.nlm.nih.gov/geo/query/acc.cgi?acc=GSE124535. Source data are provided with this paper.

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

## Acknowledgements

We thank Drs. Tian Xu and Xiaohui Wu for the *piggyBac* system; Weina Shang for assistance with EM and confocal imaging; Fangwei Wang for HeLa cells; and Junfang Ji for Huh-7 cells; Yongchao Zhao for assistance on Seahorse analysis; Chenliang Wang for

making the model art; and the core facility of the Life Sciences Institute for technical assistance. This work was supported by grants to B.Z. from the National Natural Science Foundation of China Key Project (81730069), the National Key R&D Program of China (2017YFA0504502), the National Natural Science Foundation of China General Project (31970726), Natural Science Foundation of Zhejiang key project (LZ21C070002), and the Fundamental Research Funds for the Central Universities.

## Author contributions

B.Z. and S.L. conceived the project, B.Z., S.L., and S.H. designed the study. S.L., S.H., H.Z., J.W., Y.Zhao, L.S., and L.L. performed experiments and analyzed data. Y.Zhu and C.Y. performed MS analysis of metabolites and analyzed data. J.Z. and Q.Z. analyzed data. D.Z., X.-H.F., and T.L. provided critical reagents and conceptual advice. B.Z. and S.L. wrote the manuscript with comments from all authors.

## Competing interests

The authors declare no competing interests.
