## [Peer Review File · Nature Communications]

FUNDC2 promotes liver tumorigenesis by inhibiting MFN1-mediated mitochondrial fusionREVIEWER COMMENTS

Reviewer #1 (Remarks to the Author):

Li et al. identified that mitochondrial protein FUNDC2 was found to be over-expressed in human HCC. They first showed that knockdown of FUNDC2 led to tumor repression in KRAS and cMYC over-expressing mouse HCC. Functionally, FUNDC2 was found to localize to mitochondria to promote mitochondrial fission and maintaining a robust mitochondrial respiratory efficiency. Interestingly, through proteomics, FUNDC2 was found to interact with MFN1. FUNDC2 was found to inhibit the GTPase of MFN1, a gene pivotal to mitochondrial fusion, thereby allowing HCC cells to undergo mitochondrial fission through which sustains respiration and ATP production. Taken together, this study is quite novel. Implications of FUNDC2 in HCC have not been reported previously. The role of FUNDC2 in suppressing MFN1-mediated mitochondrial fusion has never been established. Overall, the experiments were carried out properly and manuscript was written clearly. However, some important conceptual issues need to be addressed for consideration of publications. 1. Metabolic alterations induced by FUNDC2/MFN1 axis need to be more extensively studied. Currently, only oxygen consumption, ATP level, and mitochondrial lengths were evaluated. Detailed metabolic changes in the cell and mouse models have been missing and should be characterized. 2. This study lacks rescue experiment to substantiate the link between FUNDC2 and MFN1. 3. Mitochondrial fusion was believed to allow cancer cells to favor glycolysis (Warburg Effect); however, this current study oppositely suggested that mitochondrial fission favors growth of HCC cells. How authors reconcile the common understanding on tumor metabolism and their current findings.

Major concerns:

- Authors have done rescue studies to knockdown FUNDC2 and rescue with FUNDC2. However, these rescue experiments do not support the FUNDC2 regulatory link on MFN1.

(1) A FUNDC2 resistant MFN1 construct should be established to confirm the effect of mitochondrial fission is really mediated through MFN1 suppression. Since authors showed that FUNDC2 interacts with and affects MFN1 GTPase activity, authors should construct a MFN1 mutant that could not interact with FUNDC2 in the FUNDC2 OE HCC cells.

- (2) MFN1 knockdown should be done on top of FUNDC2 knockdown to confirm whether the effect of FUNDC2 depletion is abolished after MFN1 is depleted.

o In vitro (mitochondrial number counting, cell proliferation, and colony formation) and in vivo (hydrodynamic tail vein injection) assays could be validated with these rescue systems.

- Although it is clear in the field that mitochondria remain intact in cancer, ATP is considered not the major factor for sustaining growth. The building blocks and reducing (anti-oxidative) capability of cancer cells are on the other hand fundamental to cancer growth. This study's findings are interesting however do not align with the general understanding in the field. Mass spectrometry-based studies to quantify the key building blocks and more preferably metabolomics studies are necessary to characterize the overall metabolic impact of FUNDC2/MFN1 in cancer cells.

- Increased aerobic glycolysis (Warburg Effect) is a well-characterized and well-accepted cancer hallmark. However, authors only focused on studying the oxygen consumption rate of FUNDC2 KD/OE cells. It is important to learn whether increase of mitochondrial fission (respiration) led to reduction of glycolysis. If so, why cancer cells still grow faster in their models?

The following assays have to be confirmed in their FUNDC2 KD/OE HCC cells as well as the FUNDC2 OE rescued with MFN1 HCC cells or FUNDC2 KD MFN1 KD HCC cells (compared to the according controls):

1. Glucose uptake assay

2. Carbon tracing study with U-13C6 glucose to study the glycolytic flux of these cells

- Did increased mitochondrial fission/oxidative phosphorylation lead to elevation of oxidative stress as mitochondria are the places where ROS are produced. If so, why would the cells grow faster as ROS could perturb cell growth.

1. ROS quantification can be done by simple CMH2DCFDA staining followed by flow analysis
2. Mitochondrial potential has to be measured with TMRE and JC1 dye. Mitochondrial mass should be evaluated in these cells as well by NAO staining.

- Authors briefly stained Oil-Red to indicate the potential roles of FUNDC2 depletion in dysregulated lipid metabolism. Is this related to elevation of MFN1-mediated mitochondrial fusion leading to reduced lipid break down by beta oxidation? Experiments have to be performed to study the relationship of FUNDC2/MFN1 in mitochondrial fission and beta oxidation (lipid break This point needs to be clarified experimentally and literally. Would Oil-Red staining be restored when MFN-1 is knocked down in the mouse model?

- Authors also mentioned that FUNDC2/MFN-1/mitochondrial fission might promote cancer stem cells. This statement is a bit confounding. Do cancer stem cells exhibit more mitochondrial fission? Do cancer stem cells express more FUNDC2 and less MFN-1? If so, why mitochondrial fission metabolically benefits stemness maintenance and self-renewal properties of cancer stem cells? This point needs to be clarified experimentally. If this cannot be achieved, it's better not to draw any conclusion or speculations on the roles of cancer stem cells and mitochondrial fission and fusion as this would create confounding information to readers.

Reviewer #2 (Remarks to the Author):

In this manuscript the authors report that the mitochondrial protein FUNDC2 is up-regulated in human hepatocellular carcinoma and in primary mouse liver tumors, and that FUNDC2 deficiency inhibits liver tumorigenesis in mice. They also document that FUNDC2 interacts with MFN1 or MFN2 proteins, and that it inhibits MFN1 activity. Based on these observations, the authors conclude that FUNDC2 inhibition of MFN1 is a relevant process in liver carcinogenesis. The data are at odds with prior observations indicating that liver MFN1 depletion confers resistance to the development of obesity, and diabetes in response to a high fat diet (Kulkarni et al., 2015). Based on the evidence that MFN2 ablation in liver leads to a NASH-like phenotype and liver cancer in mice (Hernandez-Alvarez et al., 2019), one could think that a potential inhibitory effect of FUNDC2 on MFN2 activity would be more feasible as a pathophysiological process in liver cancer.

Major comments.

1. Based on the existing information, it is hard to think that FUNDC2-mediated MFN1 inhibition induces pathology. Thus, liver MFN1 depletion confers resistance to the development of obesity, and diabetes in response to a high fat diet (Kulkarni et al., 2015). In this connection, the authors should document that MFN1 overexpression, indeed, potentiates liver tumorigenesis, as FUNDC2 knockdown does, and that the effects of FUNDC2 knockdown are blocked upon MFN1 overexpression.

2. The model of mouse liver cancer used is not sufficiently characterized. It would be relevant to use a carcinogen such as DEN combined with a high fat diet in order to determine more appropriately the impact of FUNDC2 knockdown. In addition, it would be relevant to induce a more permanent manipulation of FUNDC2 using adeno-associated viruses. The data presented are suggestive but not fully convincing.

3. MFN1 depletion has been reported to trigger the epithelial-to-mesenchymal transition of HCC (Huang et al., 2016), and MFN1-deficient HCC cells show lower E-cadherin values and increased mesenchymal markers. These results have been confirmed by subcutaneous xenographs in mouse models and they further support the notion that MFN1 strongly decreases the metastatic potential of HCC cells. In this context, the authors should analyze in detail whether this also occur in response to FUNDC2 overexpression.

4. It is very surprising that mitochondrial respiration is reduced under conditions in which FUNDC2 deficiency leads to mitochondrial fragmentation. In general, it is recognized that mitochondrial elongation correlates with enhanced mitochondrial respiration, whereas fragmentation links to reduced mitochondrial respiration (Liesa & Shirihai 2013). This discrepancy highly suggests that FUNDC2 also mediates other effects independently of Mitofusin proteins. The authors should document that FUNDC2 overexpression or deficiency are dependent on the presence of Mitofusins.

5. FUNDC2 seems to interact with MFN1 and also with MFN2. However, FUNDC2 only inhibits GTPase activity of MFN1. It is unclear whether this is a consequence of low sensitivity of the assay, as it is known that GTPase activity is much greater in MFN1 than in MFN2. Please, document that this is not a sensitivity issue. If that is the case, genetic overexpression of FUNDC2 should only impact MFN2 KO cells but not MFN1 KO cells. The authors should document that.

6. What are the motifs within the 1-336 fragment of MFN1 that are involved in FUNDC2 binding?

Reviewer #3 (Remarks to the Author):

Li S. et al. decipher the mitochondrial dynamics related mechanism by which mitochondrial protein FUNDC2 exerts protective roles in human hepatocellular carcinoma. The proposed manuscript for publication in Nature Communications is properly written, uses appropriate technology to study the mitochondrial properties in HCC and the data support the conclusions. Although, this manuscript would be best suitable for a cancer specific audience due to the impact of the findings, I believe that such findings could shine light in novel mechanisms to understand tumor prognosis or physiology and therefore would be of interest of a broader readership such as the one Nature Communications aims at.

In brief, Li S. et al. report that FUNDC2 interacts both with the GTPase domain of Mfn1 and Mfn2 and thus inhibits mitochondrial fusion events. As a result, mitochondria respiration is down regulated and the resulting environment leads to lipid droplet accumulation. These results are of particular interest because reducing the energetic capability of a tumor environment reduces its capacity to progress and might constitute a novel therapeutic strategy for HCC.

The manuscript is suitable for publication into Nature Communications; however, I have a list of minor comments that could elevate the manuscript and could be considered prior to publication.

1. The introduction does a thorough definition of mitochondrial dynamics going over all major proteins and its functions. However, the authors miss mitochondrial fission protein Fis1. A sentence in this regard could be incorporated.

2. On page 11 authors mention that they profiled expression of FUNDC2 in 20 HCC samples by qPCR. I believe authors should provide those data and not refer to it as data not shown.

3. Have the authors consider evaluating the role between FUNDC2 and Mfn2. Authors nicely demonstrate that FUNDC2 binds to both mitofusins and although knockdown of FUNDC2 only increases binding to the GTPase domain of Mfn1 and not Mfn2, other functions or roles of Mfn2 might be impaired. For instance, Mfn2 has been extensively reported to have key roles in tethering the mitochondria to the endoplasmic reticulum. An impaired tethering to the ER could produce impairments in ER stress, ROS production and as a result in protein folding that could have also key implications for tumor progression. I acknowledge the center stage of this manuscript is the Mfn1 GTPase related roles and the authors provide a thorough characterization. However due to the interaction between Mfn2 and FUNDC2 some of these effects could be evaluated in some of the existing data. Are there less or more ER-mito contacts? Are gene profiles of ER stress impaired? At least this could be discussed.

4. When authors report the roles of FUNDC2 in mitochondrial dysregulation of tumors, leading to mitochondria fragmentation and swollen cristae they show that knockdown of FUNDC2 restores these effects. Due to the direct association between Opa1 and mitochondrial cristae, is there a link between those two proteins? Could be it be evaluated and discussed here?

5. An interesting point that is brought into the discussion is the potential to promote stemness through more efficient disposal of damaged mitochondria portions. Could that be evaluated and provide some data in those regard? Would be an interesting concept to explain the mechanism.

Overall, I want to emphasize that this is an adequate manuscript by Li S. et al. for publication in Nature Communications, following the journal's scope with only minor concerns to be addressed or discussed, proposing a novel mitochondrial therapeutic option for HCC.

REVIEWER COMMENTS

Reviewer #1 (Remarks to the Author):

Li et al. identified that mitochondrial protein FUNDC2 was found to be over-expressed in human HCC. They first showed that knockdown of FUNDC2 led to tumor repression in KRAS and cMYC over-expressing mouse HCC. Functionally, FUNDC2 was found to localize to mitochondria to promote mitochondrial fission and maintaining a robust mitochondrial respiratory efficiency. Interestingly, through proteomics, FUNDC2 was found to interact with MFN1. FUNDC2 was found to inhibit the GTPase of MFN1, a gene pivotal to mitochondrial fusion, thereby allowing HCC cells to undergo mitochondrial fission through which sustains respiration and ATP production. Taken together, this study is quite novel. Implications of FUNDC2 in HCC have not been reported previously. The role of FUNDC2 in suppressing MFN1-mediated mitochondrial fusion has never been established. Overall, the experiments were carried out properly and manuscript was written clearly. However, some important conceptual issues need to be addressed for consideration of publications.

We thank the reviewer for a positive opinion on our manuscript and giving many constructive suggestions. We further revised the manuscript according to specific comments as described below.

1. Metabolic alterations induced by FUNDC2/MFN1 axis need to be more extensively studied. Currently, only oxygen consumption, ATP level, and mitochondrial lengths were evaluated. Detailed metabolic changes in the cell and mouse models have been missing and should be characterized.

Answer:

As responded to specific comments below, we have done new experiments to determine metabolic changes downstream of FUNDC2/MFN1 due to altered mitochondrial dynamics, including MS-based targeted metabolomics, MS-based lipidomics, ¹³C₆-glucose tracer glucose flux assay, and ¹³C₁₆-palmitate tracer fatty acid flux assay, measuring ECAR by seahorse, measuring OCR when palmitate was used as energy source using seahorse, measuring mitochondrial mass, mitochondrial membrane potential, and cellular ROS levels by FACS. Results were described below.

2. This study lacks rescue experiment to substantiate the link between FUNDC2 and MFN1.

Answer:

As responded to specific comments below, we have done new experiments by knock down or knockout of *MFN1* in addition to *shFUNDC2*, and further rescue with *MFN1* wildtype or W239A mutant. Nine assays were done *in vitro* and *in vivo*, and all results support that FUNDC2 functions through MFN1.

3. Mitochondrial fusion was believed to allow cancer cells to favor glycolysis (Warburg Effect); however, this current study oppositely suggested that mitochondrial fission favors growth of HCC cells. How authors reconcile the common understanding on tumor metabolism and their current findings.

Answer:

As the reviewer pointed out, it was classically thought that mitochondrial fusion promotes respiration and glycolysis. However, it was also reported that liver-specific *MFN1* KO results in not only a highly fragmented mitochondrial network, but also enhanced mitochondrial respiration (Kulkarni et. al., *Diabetes*, 2016). This report is consistent with our model of FUNDC2 inhibition of MFN1 promotes mitochondrial fragmentation and tumorigenesis. It also suggests that the biological consequence of mitochondrial fragmentation could be context-dependent. Following the reviewer's suggestions, we performed new experiments to characterize metabolic reprogramming induced by FUNDC2-MFN1. As described below, we carried out MS-based targeted metabolomics, and found that metabolites of the TCA cycle and glycolysis (building blocks or macromolecular biosynthesis) were decreased by *FUNDC2* knockdown in tumors (Fig. 3j, Supplementary Table 1). In contrast, purine and pyrimidine metabolites were increased. Metabolism of amino acids were both increased and decreased by *FUNDC2* knockdown. Furthermore, by MS-based lipidomics, we confirmed accumulation of storage lipids commonly found in lipid droplets, including triglycerides, diglycerides, cholesteryl esters, and sterols (Fig. 3k, Supplementary Table 2). However, phospholipids which play critical roles in membrane formation and lipid signals fueling cell proliferation and malignancy, were greatly reduced by knockdown of *FUNDC2*. Furthermore, by glucose flux assay using ¹³C₆-glucose tracer, we found decreased glycolysis, TCA cycle, and increased PPP (Fig. 7d-g). Thus, by inhibiting MFN1, FUNDC2 could promote tumor growth not only by providing ATP, but also increased metabolic intermediates, and phospholipids, following dysregulated mitochondrial dynamics.

Major concerns:

- Authors have done rescue studies to knockdown FUNDC2 and rescue with FUNDC2. However, these rescue experiments do not support the FUNDC2 regulatory link on MFN1.

(1) A *FUNDC2* resistant *MFN1* construct should be established to confirm the effect of mitochondrial fission is really mediated through *MFN1* suppression. Since authors showed that *FUNDC2* interacts with and affects *MFN1* GTPase activity, authors should construct a *MFN1* mutant that could not interact with *FUNDC2* in the *FUNDC2* OE HCC cells.

Answer:

We thank reviewer's suggestion. K88T, K222Q and W239A mutations were reported to impair *MFN1* GTPase activity. We found that interactions of these mutants, especially W239A, with *FUNDC2* were largely reduced (Fig. 4h). We thus used W239A for rescue experiments in comparison to the wildtype protein as described in response to the next point (see below). Basically, all effects of *shFUNDC2* could be rescued by knockout of *MFN1*, and further reversed by expression of *MFN1* wildtype, but not the W239A *in vitro* and *in vivo*. It should be noted that W239A loses both GTPase activity and interaction with *FUNDC2*, thus the exact reason for the loss of function in W239A could not be distinguished. To further separating the two functions, new mutants (may not be possible) would be necessary, which could be done in the future.

- (2) *MFN1* knockdown should be done on top of *FUNDC2* knockdown to confirm whether the effect of *FUNDC2* depletion is abolished after *MFN1* is depleted. *In vitro* (mitochondrial number counting, cell proliferation, and colony formation) and *in vivo* (hydrodynamic tail vein injection) assays could be validated with these rescue systems.

Answer:

We thank reviewer's suggestion. In the previous manuscript, we have shown that knockdown of *MFN1* in addition to knockdown of *FUNDC2* in Huh-7 and HepG2 cells rescued the effect on mitochondria length, respiration, and ATP level. According to suggestions of the reviewer we carried new experiments showing that:

In vitro:

1. Knockout of *MFN1* rescued Huh-7 colony formation, which could be rescued by *MFN1* wildtype, but not W239A (Fig. 6a, Supplementary Fig. 7b).
2. Mitotracker staining revealed a requirement of wildtype *MFN1* (but not W239A) for the presence of elongated mitochondria by *FUNDC2* knockdown (Fig. 6b).
3. By measuring ECAR, the regulation of glycolysis by *FUNDC2* was also found depending on *MFN1* (Fig. 6g, Supplementary Fig. 8f).
4. *FUNDC2* knockdown also reduced palmitate oxidation, which was rescued by *MFN1* knockout (Fig. 6h, Supplementary Fig. 8g).
5. By targeted metabolomics, metabolites of the TCA cycle and glycolysis decreased by *shFUNDC2* were increased by knockout of *MFN1* (Fig. 7a, b, Supplementary Table 3). In contrast, pathways increased by *shFUNDC2*, including purine metabolism were decreased by further knockout of *MFN1* (Fig. 7a, c).
6. By ¹³C₆-glucose tracer glucose flux assay, and ¹³C₁₆-palmitate tracer fatty acid flux assay, reduced glycolytic flow, TCA cycle, FAO, and increased PPP by *shFUNDC2* were all depending on *MFN1* (Fig. 7d-h).

In vivo:

7. By multiplexed genome editing, knockdown of *MFN1* abolished the tumor suppressive function of *FUNDC2* knockdown. Further co-expression of *MFN1* wildtype, but not the W239A mutant rescued the tumor suppressive function of *shFUNDC2* (Fig. 8a-c).
8. By oil red staining, lipid accumulation in tumors induced by *FUNDC2* knockdown was also *MFN1*-dependent (Fig. 8d, e).
9. Expression of AFP on protein and mRNA levels was also suppressed by *FUNDC2* knockdown in an *MFN1*-dependent manner (Fig. 8d, f).

These new data strongly support our hypothesis that *FUNDC2* regulates mitochondrial dynamics, respiration, metabolic reprogramming, and promotes liver tumorigenesis by inhibiting *MFN1*.

- Although it is clear in the field that mitochondria remain intact in cancer, ATP is considered not the major factor for sustaining growth. The building blocks and reducing (anti-oxidative) capability of cancer cells are on the other hand fundamental to cancer growth. This study's findings are interesting however do not align with the general understanding in the field. Mass spectrometry-based studies to quantify the key building blocks and more preferably metabolomics studies are necessary to characterize the overall metabolic impact of *FUNDC2*/*MFN1* in cancer cells.

Answer:

We thank the reviewer for this comment. As suggested, we carried out MS-based targeted metabolomics, and found that metabolites of the TCA cycle and glycolysis (building blocks or macromolecular biosynthesis) were decreased by *FUNDC2* knockdown in tumors (Fig. 3j, Supplementary Table 1). In contrast, purine and pyrimidine metabolites were increased. Metabolism of amino acids were both increased and decreased by *FUNDC2* knockdown. Furthermore, by MS-based lipidomics, we confirmed accumulation of storage lipids commonly found in lipid droplets, including triglycerides, diglycerides, cholesteryl esters, and sterols (Fig. 3k, Supplementary Table 2). However, phospholipids which play critical roles in membrane formation

and lipid signals fueling cell proliferation and malignancy, were greatly reduced by knockdown of *FUNDC2*. Furthermore, by glucose flux assay using $^{13}\text{C}_6$ -glucose tracer, we found decreased glycolysis, TCA cycle, and increased PPP (Fig. 7d-g). Thus, by inhibiting MFN1, *FUNDC2* could promote tumor growth not only by providing ATP, but also increased metabolic intermediates, and phospholipids, following dysregulated mitochondrial dynamics.

- Increased aerobic glycolysis (Warburg Effect) is a well-characterized and well-accepted cancer hallmark. However, authors only focused on studying the oxygen consumption rate of *FUNDC2* KD/OE cells. It is important to learn whether increase of mitochondrial fission (respiration) led to reduction of glycolysis. If so, why cancer cells still grow faster in their models?

The following assays have to be confirmed in their *FUNDC2* KD/OE HCC cells as well as the *FUNDC2* OE rescued with MFN1 HCC cells or *FUNDC2* KD MFN1 KD HCC cells (compared to the according controls):

1. Glucose uptake assay

2. Carbon tracing study with U- $^{13}\text{C}_6$ glucose to study the glycolytic flux of these cells

Answer:

To answer the reviewer's question, we carried out the following experiments: First, measuring ECAR by Seahorse demonstrated that glycolysis was also reduced by *FUNDC2* knockdown (Fig. 3h, Supplementary Fig. 3j). Furthermore, the regulation of glycolysis by *FUNDC2* was also found depending on *MFN1* (Fig. 6g, Supplementary Fig. 8f). Second, using a glucose uptake assay, we demonstrated that glucose uptake was not inhibited by *FUNDC2* knockdown (Supplementary Fig. 3k). Third, by MS-based targeted metabolomics, we found that metabolites of the TCA cycle and glycolysis were decreased by *FUNDC2* knockdown in tumors (Fig. 3j, Supplementary Table 1), and were rescued by knockout of *MFN1* (Fig. 7a, b, Supplementary Table 3). Fourth, using a $^{13}\text{C}_6$ -glucose tracer, incorporation of the label was analyzed by MS. Within 24 hours of tracing, *FUNDC2* knockdown cells had lower levels of M+3 glyceraldehyde-3-phosphate, M+3 phosphoenolpyruvate, M+3 pyruvate and M+3 lactate, indicating reduced glycolytic flow to lactate, which was also *MFN1*-dependent (Fig. 7d, e). These results indicate that mitochondria fragmentation induced by *FUNDC2* through *MFN1* inhibition promotes tumorigenesis not only through enhanced mitochondrial respiration, but also involving enhanced glycolysis, which is thus consistent with the current understanding of the contribution of glycolysis to tumorigenesis.

- Did increased mitochondrial fission/oxidative phosphorylation lead to elevation of oxidative stress as mitochondria are the places where ROS are produced. If so, why would the cells grow faster as ROS could perturb cell growth.

1. ROS quantification can be done by simple CMH2DCFDA staining followed by flow analysis

2. Mitochondrial potential has to be measured with TMRE and JC1 dye. Mitochondrial mass should be evaluated in these cells as well by NAO staining.

Answer:

As the reviewer suggested, *FUNDC2* knockdown Huh-7 cells were stained by Mitotracker Green, TMRM, and CMH2DCFDA, and analyzed by FACS (Supplementary Fig. 3c-e). The results indicate that mitochondrial mass, mitochondrial membrane potential, or cellular ROS levels were not affected by *FUNDC2* knockdown. Our results support that *FUNDC2*-*MFN1* regulates metabolic reprogramming and energy production through mitochondria fragmentation to promote tumorigenesis.

- Authors briefly stained Oil-Red to indicate the potential roles of *FUNDC2* depletion in dysregulated lipid metabolism. Is this related to elevation of *MFN1*-mediated mitochondrial fusion leading to reduced lipid break down by beta oxidation? Experiments have to be performed to study the relationship of *FUNDC2*/*MFN1* in mitochondrial fission and beta oxidation (lipid break). This point needs to be clarified experimentally and literally. Would Oil-Red staining be restored when *MFN1* is knocked down in the mouse model?

Answer:

As the reviewer suggested, we further studied the relationship of *FUNDC2*/*MFN1* in mitochondrial fragmentation and fatty acid oxidation. The following new experiments were done: First, by multiplexed genome editing, knockdown of *MFN1* abolished the tumor suppressive function of *FUNDC2* knockdown (Fig. 8a-c). Further co-expression of *MFN1* wildtype, but not the W239A mutant rescued the tumor suppressive function of *shFUNDC2*. Importantly, lipid accumulation in tumors induced by *FUNDC2* knockdown was also *MFN1*-dependent (Fig. 8d, e). Second, by MS-based lipidomics, we confirmed accumulation of storage lipids commonly found in lipid droplets, including triglycerides, diglycerides, cholesteryl esters, and sterols (Fig. 3k, Supplementary Table 2). However, phospholipids which play critical roles in membrane formation and lipid signals fueling cell proliferation and malignancy, were greatly reduced by knockdown of *FUNDC2*. Third, using Seahorse, we measured mitochondrial respiration when palmitate was used as energy source. *FUNDC2* knockdown also reduced palmitate oxidation, which was rescued by *MFN1* knockout (Fig. 6h, Supplementary Fig. 8g). Fourth, targeted metabolomics revealed reduced acetylcarnitine by *FUNDC2* knockdown in an *MFN1*-dependent manner, an indicator of mitochondrial β -oxidation (Supplementary Table 1, 3). To examine changes in FAO, we fed cells with $^{13}\text{C}_{16}$ -palmitate that is channeled into the mitochondria for FAO by a carrier palmitoylcarnitine (Fig. 7d). The

TCA cycle intermediates showed an overall reduction in M+2 isotopomers, which was rescued by *MFN1* knockout (Fig. 7h). These results confirmed that *FUNDC2* promotes FAO by inhibiting *MFN1*.

- Authors also mentioned that *FUNDC2*/*MFN-1*/mitochondrial fission might promote cancer stem cells. This statement is a bit confounding. Do cancer stem cells exhibit more mitochondrial fission? Do cancer stem cells express more *FUNDC2* and less *MFN-1*? If so, why mitochondrial fission metabolically benefits stemness maintenance and self-renewal properties of cancer stem cells? This point needs to be clarified experimentally. If this cannot be achieved, it's better not to draw any conclusion or speculations on the roles of cancer stem cells and mitochondrial fission and fusion as this would create confounding information to readers.

Answer:

We thank the reviewer for this comment. In literatures, contrary to proliferating tumor cells, cancer stem cells exhibit higher dependence on oxidative phosphorylation (Sancho, P et al., *British journal of cancer*, 2016, Park, H et al., *International journal of molecular sciences*, 2020). In breast cancer, by distinguishing pre-existing and newly synthesized mitochondrial proteins using labeling technologies, it was found that upon asymmetric cell division, stem-like cells contained a greater number of 'new' mitochondria. Furthermore, interfering with *DRP1* activity abrogated asymmetric distribution of mitochondria and reduced stem-cell properties *in vitro* (Katajisto, P et al., *Science* 2015). In our data, knockdown of *FUNDC2* reduces expression of AFP, an indicator of HCC malignancy, and a marker of liver cancer stem cells. We thus thought the cancer stem cell compartment may be more susceptible to inhibition of *FUNDC2*. However, we do not have extra data to demonstrate a mechanistic link between *FUNDC2* and CSC at this stage. We thus toned down our statements in the manuscript. CSC is now only mentioned in the discussion, with a clear statement that the possibility of *FUNDC2*-induced mitochondrial fragmentation maintains cancer stem cells in HCC awaits further investigation (page 21, line 18).

Reviewer #2 (Remarks to the Author):

In this manuscript the authors report that the mitochondrial protein FUNDC2 is up-regulated in human hepatocellular carcinoma and in primary mouse liver tumors, and that FUNDC2 deficiency inhibits liver tumorigenesis in mice. They also document that FUNDC2 interacts with MFN1 or MFN2 proteins, and that it inhibits MFN1 activity. Based on these observations, the authors conclude that FUNDC2 inhibition of MFN1 is a relevant process in liver carcinogenesis. The data are at odds with prior observations indicating that liver MFN1 depletion confers resistance to the development of obesity, and diabetes in response to a high fat diet (Kulkarni et al., 2015). Based on the evidence that MFN2 ablation in liver leads to a NASH-like phenotype and liver cancer in mice (Hernandez-Alvarez et al., 2019), one could think that a potential inhibitory effect of FUNDC2 on MFN2 activity would be more feasible as a pathophysiological process in liver cancer.

We thank the reviewer for evaluating our manuscript and giving many constructive suggestions. Regarding the relationship between FUNDC2 and MFN1/2, it was reported that liver-specific knockout of *FUNDC2* promotes cellular accumulation of triglycerides and non-alcohol fatty liver disease (NAFLD), as well as glucose intolerance induced by high-fat diet in mice (Lu, H. et al., *Biomedecine & pharmacotherapie*, 2020). Such a phenotype is consistent with our finding of reduced FAO and accumulation of lipid droplets upon ablation of *FUNDC2*, although we proposed a completely different mechanism. In our model FUNDC2 promotes mitochondrial fragmentation and mitochondrial respiration through inhibition of MFN1, which was supported by extensive new data as described below in response to specific points. It should be noted that liver-specific knockout of *MFN1* results in not only a highly fragmented mitochondrial network, but also enhanced mitochondrial respiration (Kulkarni, S. S et al., *Diabetes*, 2016). This phenotype is thus similar to FUNDC2 activation in liver tumor cells, and is consistent with our model of MFN1 inhibition by FUNDC2. In this report it was demonstrated that *MFN1* deficiency increased complex I abundance, which thus may explain increased respiration. Thus, although mitochondrial fragmentation was more thought to reduce respiration, that caused by inhibition of MFN1 seems different. Whether this difference is due to other functions of MFN1 is worth further investigation. We have also provided new data excluding MFN2 as a downstream effector of FUNDC2 as responded below to point 5. We respond to the reviewer's specific comments as below.

Major comments.

1. Based on the existing information, it is hard to think that FUNDC2-mediated MFN1 inhibition induces pathology. Thus, liver MFN1 depletion confers resistance to the development of obesity, and diabetes in response to a high fat diet (Kulkarni et al., 2015). In this connection, the authors should document that MFN1 overexpression, indeed, potentiates liver tumorigenesis, as FUNDC2 knockdown does, and that the effects of FUNDC2 knockdown are blocked upon MFN1 overexpression.

Answer:

We thank the reviewer's suggestion. As suggested, we performed new *in vivo* experiments demonstrating that co-expression of *MFN1* with *MYC+RAS* strongly suppressed tumorigenesis (Supplementary Fig. 9a-c). [Please note that *FUNDC2* knockdown suppress instead of promote tumorigenesis.] This result is in consistent with that inhibition of MFN1 is a key mechanism of tumor promotion by FUNDC2.

In the previous manuscript, we have shown that knockdown of MFN1 in addition to knockdown of FUNDC2 in Huh-7 and HepG2 cells rescued the effect on mitochondria length, respiration, and ATP level. According to suggestions of the reviewer we carried new experiments showing that:

In vitro:

1. Knockout of *MFN1* rescued Huh-7 colony formation, which could be rescued by MFN1 wildtype, but not W239A (Fig. 6a, Supplementary Fig. 7b).
2. Mitotracker staining revealed a requirement of wildtype MFN1 (but not W239A) for the presence of elongated mitochondria by *FUNDC2* knockdown (Fig. 6b).
3. By measuring ECAR, the regulation of glycolysis by *FUNDC2* was also found depending on *MFN1* (Fig. 6g, Supplementary Fig. 8f).
4. *FUNDC2* knockdown also reduced palmitate oxidation, which was rescued by *MFN1* knockout (Fig. 6h, Supplementary Fig. 8g).
5. By targeted metabolomics, metabolites of the TCA cycle and glycolysis decreased by *shFUNDC2* were increased by knockout of *MFN1* (Fig. 7a, b, Supplementary Table 3). In contrast, pathways increased by *shFUNDC2*, including purine metabolism were decreased by further knockout of *MFN1* (Fig. 7a, c).
6. By ¹³C₆-glucose tracer glucose flux assay, and ¹³C₁₆-palmitate tracer fatty acid flux assay, reduced glycolytic flow, TCA cycle, FAO, and increased PPP by *shFUNDC2* were all depending on *MFN1* (Fig. 7d-h).

In vivo:

7. By multiplexed genome editing, knockdown of *MFN1* abolished the tumor suppressive function of *FUNDC2* knockdown. Further co-expression of *MFN1* wildtype, but not the W239A mutant rescued the tumor suppressive function of *shFUNDC2* (Fig. 8a-c).

8. By oil red staining, lipid accumulation in tumors induced by *FUNDC2* knockdown was also *MFN1*-dependent (Fig. 8d, e).

9. Expression of AFP on protein and mRNA levels was also suppressed by *FUNDC2* knockdown in an *MFN1*-dependent manner (Fig. 8d, f).

These new data strongly support our hypothesis that *FUNDC2* regulates mitochondrial dynamics, respiration, metabolic reprogramming, and promotes liver tumorigenesis by inhibiting *MFN1*.

2. The model of mouse liver cancer used is not sufficiently characterized. It would be relevant to use a carcinogen such as DEN combined with a high fat diet in order to determine more appropriately the impact of *FUNDC2* knockdown. In addition, it would be relevant to induce a more permanent manipulation of *FUNDC2* using adeno-associated viruses. The data presented are suggestive but not fully convincing.

Answer:

We appreciate the reviewer's discussion on relevance of models for investigating *FUNDC2*-*MFN1* functions. We had considered DEN-induced liver tumorigenesis model. However, there are a few disadvantages using this model. As previously reported, by genomic and transcriptomic profiling, DEN-induced mouse liver tumors are least similar to the human disease among four models examined (Dow, et. al., *PNAS*, 2018), although it is widely used. Second, in order to functionally characterize a gene of interest in carcinogen-induced models like DEN, tissue-specific KO or transgenic lines are necessary. However, for genes involved in physiological functions in addition to tumorigenesis, the phenotype could be difficult to interpret. For example, liver-specific knockout of *FUNDC2* was reported to induce cellular accumulation of triglycerides and non-alcohol fatty liver disease (NAFLD), as well as glucose intolerance induced by high-fat diet in mice (Lu, H. et al., *Biomedicine & pharmacotherapy*, 2020). In addition, liver-specific knockout of *MFN1* increases mitochondrial content and respiration, promotes the use of lipids as energy source, and protects the mice against high-fat diet-induced glucose intolerance (Kulkarni, S. S et al., *Diabetes*, 2016). Thus, both *FUNDC2* and *MFN1* KO exhibit systematic effect, which will affect non-tumor tissue, thus could result in indirect effects. In addition, the DEN model takes a long time to induce tumorigenesis.

Mouse liver tumors induced by *MYC+RAS* using traditional genetic methods (Xu, et. al., *Nature Med*, 2019) or hydrodynamic injection of transposons (Ju, et. al., *Int J Cancer*, 2016; Xin, et. al., *Oncogene*, 2017; Tipanee, et. al., *Molecular Therapy: Nucleic Acids*, 2020) were previously reported. By pathological analysis and molecular profiling, mouse liver tumors induced by hydrodynamic injection of *MYC+RAS* transposons were reported to be similar to the steatohepatitic subtype of human HCC (Tipanee, et. al., *Molecular Therapy: Nucleic Acids*, 2020). We confirmed this finding in our model (new Supplementary Fig. 1a). We used hydrodynamic injection-based multiplexed *in situ* genome editing of mouse hepatocytes to investigate *FUNDC2*-*MFN1*. This method was recently recognized by the liver cancer field as a useful tool, and was used by increasing studies (Xue, *Nature*, 2014; Seehawer, *Nature*, 2018; Molina-Sanchez, *Gastro*, 2020; reviewed by Xin Chen, *Am J of Path*, 2014). The mosaic nature of genome editing through hydrodynamic injection allows silencing of target genes precisely in tumor cells, leaving the para-tumor tissue intact. Thus, the tumor-specific roles of *FUNDC2*-*MFN1* could be determined without interference from disturbed para-tumor tissues. It should be noted that modification of the genome by this method is permanent since ORFs were integrated into the genome by transient expression of the transposase.

3. *MFN1* depletion has been reported to trigger the epithelial-to-mesenchymal transition of HCC (Huang et al., 2016), and *MFN1*-deficient HCC cells show lower E-cadherin values and increased mesenchymal markers. These results have been confirmed by subcutaneous xenographs in mouse models and they further support the notion that *MFN1* strongly decreases the metastatic potential of HCC cells. In this context, the authors should analyze in detail whether this also occur in response to *FUNDC2* overexpression.

Answer:

To answer the reviewer's question, we analyzed EMT signatures in differential genes comparing control and *FUNDC2* knockdown tumors. Gene Set Enrichment Analysis (GSEA) indicated that EMT signatures were not enriched in *FUNDC2*-regulated genes (Figure 1 for reviewer).

Figure 1 for reviewer. Genes regulated by *FUNDC2* in *MYC+RAS* tumors were subjected to GSEA analysis.

4. It is very surprising that mitochondrial respiration is reduced under conditions in which FUNDC2 deficiency leads to mitochondrial fragmentation. In general, it is recognized that mitochondrial elongation correlates with enhanced mitochondrial respiration, whereas fragmentation links to reduced mitochondrial respiration (Liesa & Shirihai 2013). This discrepancy highly suggests that FUNDC2 also mediates other effects independently of Mitofusin proteins. The authors should document that FUNDC2 overexpression or deficiency are dependent on the presence of Mitofusins.

Answer:

As we responded to point 1, according to previous and new experiments performed during the revision, all effects induced by *FUNDC2* knockdown are dependent on MFN1. These results support our hypothesis that *FUNDC2* regulates mitochondrial dynamics, respiration, metabolic reprogramming, and promotes liver tumorigenesis by inhibiting MFN1. In a previous report, by generating liver-specific *MFN1* LKO mice, it was reported that *MFN1* KO results in not only a highly fragmented mitochondrial network, but also enhanced mitochondrial respiration (Kulkarni et al., *Diabetes*, 2016). Here it was demonstrated that *MFN1* deficiency increased complex I abundance, which thus may explain increased respiration. Thus, although mitochondrial fragmentation was more thought to reduce respiration, that caused by inhibition of MFN1 seems different. Whether this difference is due to other functions of MFN1 is worth further investigation.

5. FUNDC2 seems to interact with MFN1 and also with MFN2. However, FUNDC2 only inhibits GTPase activity of MFN1. It is unclear whether this is a consequence of low sensitivity of the assay, as it is known that GTPase activity is much greater in MFN1 than in MFN2. Please, document that this is not a sensitivity issue. If that is the case, genetic overexpression of FUNDC2 should only impact MFN2 KO cells but not MFN1 KO cells. The authors should document that.

Answer:

In Fig. 5a, MFN1 and MFN2 were expressed in HEK293T cells, and immunoprecipitated for activity assay. Previously, we separately normalized MFN1/2+*FUNDC2* groups to MFN1/2 without *FUNDC2*, so that the effect of *FUNDC2* was clear, but the difference between MFN1 and MFN2 could not be judged. In the revised manuscript, we adjusted our normalization using the group MFN1 without *FUNDC2* as unit one, it is now obvious that MFN2 activity is only mildly lower. It should be noted that we transfected five times more plasmids for MFN2, knowing that MFN2 activity is lower than MFN1. Under this condition, we could conclude that inhibition of MFN1 but not MFN2 by *FUNDC2* was not due to a sensitivity issue.

Furthermore, as the reviewer suggested, we generated *MFN1* or *MFN2* knockout Huh-7 cells by CRISPR/Cas9, and cells were further infected for knockdown of *FUNDC2* (Supplementary Fig. 7a). Colony formation assay indicated that knockout of *MFN1* but not *MFN2* eliminated the effect of *FUNDC2* knockdown on reducing colony formation (Fig. 6a, Supplementary Fig. 7b). Furthermore, Mitotracker staining revealed a requirement of wildtype MFN1, but not MFN2 for the presence of elongated mitochondria by *FUNDC2* knockdown (Fig. 6b). In addition, the effects of *MFN1* knockout could be rescued by re-expression of wild type *MFN1*.

MFN2 was also reported to play a key role in tethering mitochondria to endoplasmic reticulum (ER), and impaired tethering could result in ER stress (Sebastian, D et al., *PNAS*, 2012). We thus quantified mitochondria associated ER-membranes (MAM) in tumors, and found that the percentage of MAM to mitochondria perimeter was not affected by *FUNDC2* knockdown (Supplementary Fig. 7c). In addition, knockdown of *FUNDC2* in Huh-7 cells or tumors did not induce ER stress as indicated by phosphorylation level of PERK and protein levels of Bip and ATF4 (Supplementary Fig. 7d, e).

These results demonstrate that *FUNDC2* functions through inhibition of MFN1, but not MFN2.

6. What are the motifs within the 1-336 fragment of MFN1 that are involved in FUNDC2 binding?

Answer:

The 1-336 fragment of MFN1 is largely composed of the GTPase domain. Although four motifs G1-G4 could be distinguished by crystal structures, deletion of any of them could result in collapse of the domain structure thus would be uninformative. However, by point mutations of the GTPase domain, we did find that the W239A mutant has minimal interaction with *FUNDC2* (Fig. 4h). This mutant was also used for functional studies (Fig. 6a, 6b, 6f, 6g, 6h, 8b-8f).

Reviewer #3 (Remarks to the Author):

Li S. et al. decipher the mitochondrial dynamics related mechanism by which mitochondrial protein FUNDC2 exerts protective roles in human hepatocellular carcinoma. The proposed manuscript for publication in Nature Communications is properly written, uses appropriate technology to study the mitochondrial properties in HCC and the data support the conclusions. Although, this manuscript would be best suitable for a cancer specific audience due to the impact of the findings, I believe that such findings could shine light in novel mechanisms to understand tumor prognosis or physiology and therefore would be of interest of a broader readership such as the one Nature Communications aims at.

In brief, Li S. et al. report that FUNDC2 interacts both with the GTPase domain of Mfn1 and Mfn2 and thus inhibits mitochondrial fusion events. As a result, mitochondria respiration is down regulated and the resulting environment leads to lipid droplet accumulation. These results are of particular interest because reducing the energetic capability of a tumor environment reduces its capacity to progress and might constitute a novel therapeutic strategy for HCC.

The manuscript is suitable for publication into Nature Communications; however, I have a list of minor comments that could elevate the manuscript and could be considered prior to publication.

We thank the reviewer for a positive opinion on our manuscript. We also appreciate the constructive suggestions. We further revised the manuscript according to the specific comments as described below.

1. The introduction does a thorough definition of mitochondrial dynamics going over all major proteins and its functions. However, the authors miss mitochondrial fission protein Fis1. A sentence in this regard could be incorporated.

Answer:

We thank the reviewer for this suggestion. An introduction of the roles of FIS1 in mitochondrial dynamics was added in the revised manuscript (page 3, line 18).

2. On page 11 authors mention that they profiled expression of FUNDC2 in 20 HCC samples by qPCR. I believe authors should provide those data and not refer to it as data not shown.

Answer:

We thank the reviewer's suggestion. The qPCR result is now provided in Supplementary Fig. 6c.

3. Have the authors consider evaluating the role between FUNDC2 and Mfn2. Authors nicely demonstrate that FUNDC2 binds to both mitofusins and although knockdown of FUNDC2 only increases binding to the GTPase domain of Mfn1 and not Mfn2, other functions or roles of Mfn2 might be impaired. For instance, Mfn2 has been extensively reported to have key roles in tethering the mitochondria to the endoplasmic reticulum. An impaired tethering to the ER could produce impairments in ER stress, ROS production and as a result in protein folding that could have also key implications for tumor progression. I acknowledge the center stage of this manuscript is the Mfn1 GTPase related roles and the authors provide a thorough characterization. However due to the interaction between Mfn2 and FUNDC2 some of these effects could be evaluated in some of the existing data. Are there less or more ER-mito contacts? Are gene profiles of ER stress impaired? At least this could be discussed.

Answer:

As the reviewer suggested, we further examined the potential roles of MFN1 and MFN2 downstream of FUNDC2. We generated *MFN1* or *MFN2* knockout Huh-7 cells by CRISPR/Cas9, and cells were further infected for knockdown of *FUNDC2* (Supplementary Fig. 7a). Colony formation assay indicated that knockout of *MFN1* but not *MFN2* eliminated the effect of *FUNDC2* knockdown on reducing colony formation (Fig. 6a, Supplementary Fig. 7b). Furthermore, Mitotracker staining revealed a requirement of wildtype MFN1, but not MFN2 for the presence of elongated mitochondria by *FUNDC2* knockdown (Fig. 6b). In addition, the effects of *MFN1* knockout could be rescued by re-expression of wild type *MFN1*.

MFN2 was also reported to play a key role in tethering mitochondria to endoplasmic reticulum (ER), and impaired tethering could result in ER stress. We thus quantified mitochondria associated ER-membranes (MAM) in tumors, and found that the percentage of MAM to mitochondria perimeter was not affected by *FUNDC2* knockdown (Supplementary Fig. 7c). In addition, knockdown of *FUNDC2* in Huh-7 cells or tumors did not induce ER stress as indicated by phosphorylation level of PERK and protein levels of Bip and ATF4 (Supplementary Fig. 7d, e).

These results further demonstrate that FUNDC2 functions through inhibition of MFN1, but not MFN2.

4. When authors report the roles of FUNDC2 in mitochondrial dysregulation of tumors, leading to mitochondria fragmentation and swollen cristae they show that knockdown of FUNDC2 restores these effects. Due to the direct association between Opa1 and mitochondrial cristae, is there a link between those two proteins? Could be it be evaluated and discussed here?

Answer:

As the reviewer suggested, we examined the interaction between FUNDC2 and OPA1 by co-immunoprecipitation, and no interaction could be detected, although MFN1 as a positive control could co-IP with FUNDC2 (Figure 2 for reviewer).

Figure 2 for reviewer. HEK293T cells were transfected and lysates were immunoprecipitated with anti-HA antibody.

5. An interesting point that is brought into the discussion is the potential to promote stemness through more efficient disposal of damaged mitochondria portions. Could that be evaluated and provide some data in those regard? Would be an interesting concept to explain the mechanism.

Answer:

We thank the reviewer for this comment. In our data, knockdown of *FUNDC2* reduces expression of AFP, an indicator of HCC malignancy, and a marker of liver cancer stem cells. In literatures, contrary to proliferating tumor cells, cancer stem cells exhibit higher dependence on oxidative phosphorylation (Sancho, P et al., *British journal of cancer*, 2016, Park, H et al., *International journal of molecular sciences*, 2020). In breast cancer, by distinguishing pre-existing and newly synthesized mitochondrial proteins using labeling technologies, it was found that upon asymmetric cell division, stem-like cells contained a greater number of 'new' mitochondria. Furthermore, interfering with DRP1 activity abrogated asymmetric distribution of mitochondria and reduced stem-cell properties *in vitro* (Katajisto, P et al., *Science* 2015). We thus thought the cancer stem cell compartment may be more susceptible to inhibition of FUNDC2. However, we do not have extra data to demonstrate a mechanistic link between FUNDC2 and CSC at this stage. As reviewer 1 suggested, we toned down our statements in the manuscript. CSC is now only mentioned in the discussion, with a clear statement that the possibility of FUNDC2-induced mitochondrial fragmentation maintains cancer stem cells in HCC awaits further investigation (page 21, line 18).

Overall, I want to emphasize that this is an adequate manuscript by Li S. et al. for publication in Nature Communications, following the journal's scope with only minor concerns to be addressed or discussed, proposing a novel mitochondrial therapeutic option for HCC.

REVIEWERS' COMMENTS

Reviewer #1 (Remarks to the Author):

The authors have adequately addressed my questions. I have no further comments.

Reviewer #2 (Remarks to the Author):

None

Reviewer #3 (Remarks to the Author):

All my concerns have been addressed and authors have conducted a thorough answer to all of reviewers comments. Given the interest of the paper and the quality of the manuscript I believe no additional concerns should be considered.